# Mre11-Rad50 oligomerization promotes DNA double-strand break repair

Vera M. Kissling [1], Giordano Reginato[1,2,8], Eliana Bianco [1,8], Kristina Kasaciunaite[3], Janny Tilma[1], Gea Cereghetti [1], Natalie Schindler[4], Sung Sik Lee[1,5], Raphaël Guérois [6], Brian Luke[4,7], Ralf Seidel [3], Petr Cejka[1,2✉] & Matthias Peter [1✉]

The conserved Mre11-Rad50 complex is crucial for the detection, signaling, end tethering and processing of DNA double-strand breaks. While it is known that Mre11-Rad50 foci formation at DNA lesions accompanies repair, the underlying molecular assembly mechanisms and functional implications remained unclear. Combining pathway reconstitution in electron microscopy, biochemical assays and genetic studies, we show that *S. cerevisiae* Mre11-Rad50 with or without Xrs2 forms higher-order assemblies in solution and on DNA. Rad50 mediates such oligomerization, and mutations in a conserved Rad50 beta-sheet enhance or disrupt oligomerization. We demonstrate that Mre11-Rad50-Xrs2 oligomerization facilitates foci formation, DNA damage signaling, repair, and telomere maintenance in vivo. Mre11-Rad50 oligomerization does not affect its exonuclease activity but drives endonucleolytic cleavage at multiple sites on the 5'-DNA strand near double-strand breaks. Interestingly, mutations in the human RAD50 beta-sheet are linked to hereditary cancer predisposition and our findings might provide insights into their potential role in chemoresistance.

[1] Department of Biology, Institute of Biochemistry, Eidgenössische Technische Hochschule (ETH), 8093 Zürich, Switzerland. [2] Institute for Research in Biomedicine, Università della Svizzera italiana (USI), Faculty of Biomedical Sciences, 6500 Bellinzona, Switzerland. [3] Peter Debye Institute for Soft Matter Physics, Universität Leipzig, 04103 Leipzig, Germany. [4] Institute for Developmental and Neurobiology (IDN), Johannes Gutenberg University, 55128 Mainz, Germany. [5] Scientific Center for Optical and Electron Microscopy, Eidgenössische Technische Hochschule (ETH), 8093 Zürich, Switzerland. [6] Institute for Integrative Biology of the Cell (I2BC), Commissariat à l'Energie Atomique, CNRS, Université Paris-Sud, Université Paris-Saclay, 91190 Gif-sur-Yvette, France. [7] Institute of Molecular Biology (IMB), 55128 Mainz, Germany. [8]These authors contributed equally: Giordano Reginato, Eliana Bianco. ✉email: petr.cejka@irb.usi.ch; matthias.peter@bc.biol.ethz.ch

Misfunction of the MRE11-RAD50-NBS1 (MRN) complex is associated with several pathologies, including various hereditary and sporadic cancers[1]. Aberrant degradation of nascent DNA by MRE11 was observed in tumor suppressor BRCA1/2-deficient cells, and abrogation of MRE11 activity is linked to PARP inhibitor resistance in cancer chemotherapy[2,3]. At the molecular level, human MRN and yeast Mre11-Rad50-Xrs2 (MRX) promote DNA double-strand break (DSB) repair via non-homologous end joining (NHEJ) and homologous recombination (HR) by means of DNA tethering, activation of downstream pathway components, and ATP-dependent resection of the 5′-terminated strands at DSBs[4]. The latter function is particularly relevant for HR and a variant of the end-joining pathway termed alternative end-joining (A-EJ)[4]. The MRN/X complex also activates the DNA damage checkpoint kinase ATM/Tel1, which in turn phosphorylates targets to orchestrate cell cycle arrest and DNA repair[5]. Moreover, MRN/X and ATM/Tel1 contribute to the maintenance of normal telomere length[5].

Nucleolytic processing of DSBs by MRN/X is restricted to the vicinity of the breaks, and hence termed short-range DNA end resection[4,6,7]. The activity of MRN/X is versatile, as it can process DSBs with various secondary DNA structures and protein blocks, both covalently bound, such as topoisomerase cleavage complexes, or non-covalently associated, such as the NHEJ factor Ku[8–11]. This preference for cleaving protein-blocked DNA ends by MR homologs is conserved in evolution from archaea and bacteria to yeast and humans[12–15].

In budding yeast S. cerevisiae, Mre11 nuclease activity is essential for processing protein-blocked DNA ends. This activity can be bypassed in case of free DNA ends by long-range nucleases normally acting downstream, including Exo1 and Dna2[16]. NBS1/Xrs2 is essential for resection in humans, but largely dispensable in yeast. It is, however, needed for Tel1 activation and nuclear import of the MRX complex[15,17]. Due to the requirement of a sister chromatid as homologous template for repair via HR, resection is limited to the S/G2 phase of the cell cycle, which is achieved through cell cycle-regulated stimulation of Mre11 endonuclease activity by phosphorylated Sae2 (pSae2)[18,19]. In vitro, MR first incises the 5′-terminated strand ~15–35 bp away from the DSB end with its endonuclease function, and then exonucleolytically degrades DNA backwards to the end in 3′−5′ direction[9,12,15]. In cells, Mre11-dependent resection can generate 3′-overhangs of up to ~300 nt in length[20]. It is unclear whether the short-range resection tracts are generated by DNA cleavage that proceeds stepwise away from the DNA end, as observed in vitro[21], or whether MR in most cases first cleaves DNA at greater distances from the DNA end, as observed in meiotic cells[20]. The single-stranded overhangs are then elongated to more than 1000 nt by the long-range resection machinery including Exo1 or Dna2-Sgs1 for homology-directed repair[7,20,22].

Upon DNA damage in eukaryotic cells, MR relocates from a homogeneous, nuclear distribution to discrete foci consisting of hundreds to thousands of MR molecules[23–28]. This localized MR accumulation then recruits a cascade of DNA repair and signaling proteins, including ATM/Tel1[25,26]. Given that MR foci formation at DSBs and the resultant array of protein interactions are conserved in eukaryotes, the capacity of MR to quickly accumulate at DSBs in a regulated manner appears to be an essential characteristic of efficient DSB repair[25].

MR was shown to dimerize to $M_2R_2$ complexes with one or two subunits of NBS1/Xrs2, whereby the Mre11 nuclease and Rad50 ATPase domains form a globular head from which two Rad50 coiled-coils protrude, culminating at a zinc hook at the apex of the structure[29–32]. DNA binding occurs mainly via the head domain of Mre11 and Rad50, but Xrs2 and the Rad50 coiled-coils were also shown to bind DNA[27,29,33,34]. Foci formation could be facilitated by higher-order oligomerization of MR dimers [$(M_2R_2)_n$] via globular head domain interactions or by zinc hook-mediated tethering of the protruding Rad50 coiled-coils[35,36]. Although hints of head domain oligomerization have been observed for various species in vitro[27,35,37], especially for eukaryotic MR, coiled-coil tethering was predominantly investigated and functionally associated with DNA movement, condensation, and end fixation in repair[27,35,36]. While high-resolution structures of the MR dimer[34,36,38–41] from crystallography and cryo-electron microscopy contributed significantly to the understanding of the diverse MR functions, they provided only limited information on more heterogeneous higher-order structures formed by MR dimers at DSBs. Thus, the relevance of MR head domain interactions for higher-order oligomerization and for the appearance of clearly detectable MR foci upon DNA damage remains unclear. Taken together, defining the molecular mechanisms of foci assembly and identifying the interaction interfaces underlying MR oligomerization are of great importance for understanding MR-mediated functions and mechanisms in DNA damage signaling and repair.

Here we examine DNA end resection reactions by combining biochemical reconstitution and negative staining transmission electron microscopy (TEM), and thereby provide insights into the function and architecture of active DNA end resection complexes at the single molecule level. We demonstrate that short-range DNA end resection is driven by MR oligomerization mediated by MR head domain interactions, which stabilize MR on DNA and enhance its endonuclease activity at multiple sites along the 5′-terminated DNA strand. Mutational analysis shows that a small interface in the globular part of Rad50 modulates MR oligomerization and DNA end resection, as well as MR-dependent foci formation, DNA damage signaling, and telomere maintenance in S. cerevisiae. Together, these results identify MR oligomerization as a physiologically relevant process that is pre-requisite for efficient DSB repair and suggest a molecular mechanism for foci formation at DSBs.

## Results

**MR oligomerizes via head domain and forms nucleolytically active assemblies on DNA.** To understand the role and regulation of MR(X) oligomerization in DSB repair, we purified recombinant S. cerevisiae MR(X) from insect cells and visualized it with negative staining TEM (Fig. 1a and Supplementary Fig. 1a, b). Since Xrs2 is largely dispensable for DNA end resection[17] and integrated with variable copy numbers into purified MRX complexes, we mainly analyzed MR to reduce complexity. In accord with previous studies of MR homologs[29,35,36,38], we observed the formation of MR dimers ($M_2R_2$) with a globular head domain of ~11 nm in diameter and protruding ~50 nm long Rad50 coiled-coils. The coiled-coils were either open (Fig. 1a-i, ii and Supplementary Fig. 1b-i) or closed to a clamp via zinc hooks at the apex of the structure (Fig. 1a-iii and Supplementary Fig. 1b-ii). Additionally, $M_2R_2$ also displayed the capacity to oligomerize via its head domain, allowing the formation of tetramers ($M_2R_2)_2$ and larger assemblies ($M_2R_2)_n$ with protruding coiled-coils that could then further interact via tethering (Fig. 1a-ii–iv; Supplementary Fig. 1b-i–iii). ATP binding stimulates MR association with DNA[41,42], and MR endonuclease activity is magnesium- and manganese-dependent and requires ATP hydrolysis by Rad50[12,19]. Accordingly, under more physiological conditions as used for in vitro MR nuclease assays (30 °C with $Mg^{2+}$, $Mn^{2+}$, but ATPγS instead of ATP to prevent ATP hydrolysis and stabilize Rad50 interactions), the MR oligomers were even more evident (Fig. 1a-v and Supplementary Fig. 1b-iv). Higher temperature

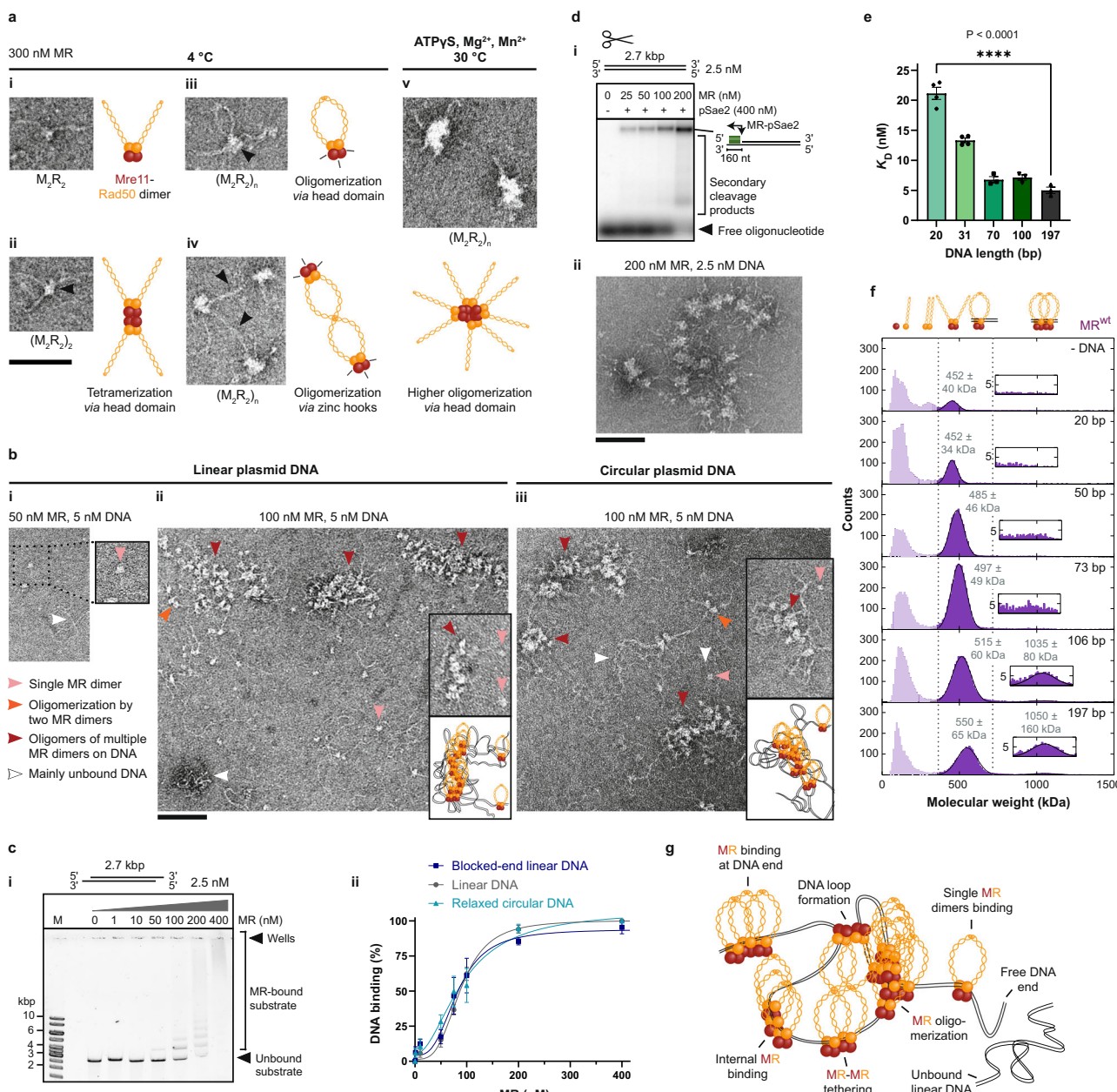

**Fig. 1 MR oligomerizes *via* head domain and forms nucleolytically active assemblies on DNA. a** TEM of *S. cerevisiae* Mre11-Rad50 dimers ($M_2R_2$) and oligomers ($M_2R_2$)$_n$ with open (i, ii) or closed (iii) Rad50 coiled-coils. MR oligomerization *via* head domains (arrows in ii, iii; gray cartoon lines in iii, iv) and further interaction through coiled-coil tethering (arrows in iv). (v) Larger MR oligomers at 30 °C with co-factors ATPγS, $Mg^{2+}$, and $Mn^{2+}$ (Supplementary Fig. 1a–c). Images representative of $n = 3$. **b** TEM of MR binding to plasmid DNA at 30 °C with co-factors as above. (i) $M_2R_2$ binding the DNA end. (ii, iii) MR dimers oligomerizing on DNA and some MR molecules forming "pearls-on-a-string"-like structures (zoom-insets of examples with cartoons), irrespective of DNA ends (Supplementary Fig. 1e, f). $n = 3$. **c** EMSA of MR binding to plasmid DNA (i) and quantification (ii; Supplementary Fig. 1g). Mean ± SEM, $n = 3$ or $n = 4$ independent experiments, sigmoidal fit. **d** (i) Resection assay with probe (green box in cartoon) binding to the 3'-overhang produced. $n = 2$. (ii) TEM of (i) without pSae2 but with ATPγS to stabilize MR on DNA. $n = 3$. Large MR oligomers still catalyze resection with pSae2 and ATP. **e** Quantification of EMSAs as in Supplementary Fig. 2a with oligonucleotides of different lengths (100 nM base pairs). $K_D$ as MR concentration at 50% DNA binding ± SEM of $n_{20,31bp} = 4$, $n_{70,100,197bp} = 3$, unpaired two-tailed *t*-test. **f** Molecular mass distributions from mass photometry show better $MR^{wt}$ binding to longer oligonucleotides (100 nM base pairs). Cartoons illustrate species in peaks. Measured molecular weight (MW, kDa) ± SD in gray. Zoom insets show MW range where a peak of two MR dimers bound to one DNA molecule would be expected and was fitted where possible (Supplementary Fig. 3c). $n = 3$ independent experiments. **g** Model illustrating that in the presence of DNA, MR oligomers dissociate to dimers which then oligomerize again *via* head domain interactions at DNA ends or internally to "pearls-on-a-string"-like assemblies and larger clusters. Substrate compaction suggests DNA loop formation and coiled-coil tethering. Scale bars: 100 nm (**a**, **b**, **d**).

facilitated the formation of larger MR oligomers (compare 4 °C in Supplementary Fig. 1b-i–iii, 30 °C in Supplementary Fig. 1b-iv and 42 °C in Supplementary Fig. 1d-i). These MR oligomers were stable at physiological salt concentrations (150 mM KCl or NaCl) and only dissociated into MR dimers ($M_2R_2$) at high salt (260 mM KCl) (Supplementary Fig. 1c). Oligomer size decreased linearly with increasing KCl concentration, suggesting a charge screening effect of salt on electrostatic interactions between the MR head domains (100 nM MR, Supplementary Fig. 1c). However, higher temperature (42 °C) and increased MR concentration (600 nM) could overcome this charge screening effect at high salt conditions, restoring MR oligomerization (compare 260 mM KCl in Supplementary Fig. 1c-i with Supplementary Fig. 1d-ii). Together, the high local MR concentrations at DSBs and the growth temperature of yeast may thus counteract the charge screening effect of chromatin DNA[43], and allow MR oligomerization under physiological conditions.

To examine the effect of DNA on MR oligomerization, we investigated MR assembly on linear and circular plasmid DNA. At lower MR:DNA ratios (50–100 nM MR, 10–20:1 with respect to DNA molecules), the MR oligomers loaded onto DNA and dissociated into dimers that bound DNA ends (Fig. 1b-i) or internally (Fig. 1b-ii, iii and Supplementary Fig. 1e, f). A few isolated MR dimers or pairs of MR dimers bound adjacently on DNA were visible (Fig. 1b, pink and orange arrows denote examples). The majority of MR molecules, however, were non-randomly distributed and oligomerized on DNA (Fig. 1b-ii, iii; red arrows). Some of these MR molecules appeared to form "pearls-on-a-string"-like assemblies on DNA, seemingly interacting with each other through the head domains, oftentimes with DNA loops in between (Fig. 1b-ii, iii zoom-insets and cartoons). The dense accumulation of MR units at some DNA regions contrasted with other DNA stretches that were left almost protein-free, as observed both in the presence and absence of DNA ends with ATP or ATPγS (Fig. 1b-ii, iii and Supplementary Fig. 1e, f). The detected DNA loops and substrate compaction suggest that the head domain interactions and subsequent Rad50 tethering of MR oligomers can bring distant parts of DNA into close proximity, as reported previously for archaeal MR[27]. In electrophoretic mobility shift assays (EMSAs) we observed a distinct banded pattern at low MR concentrations (up to 100 nM), followed by a rapid shift of the protein-DNA complexes to the wells in a narrow MR concentration range (from 200 to 400 nM) (Fig. 1c-i and Supplementary Fig. 1g). This switch-like binding behavior of MR was comparable on linear and circular DNA, irrespective of free or protein-blocked ends or DNA supercoiling (Fig. 1c-ii and Supplementary Fig. 1g), and was suggestive of a stabilization of MR units bound to DNA as a protein cluster. Limited binding preference of MR to DNA ends was observed previously, as Xrs2, albeit not essential for resection in vivo, facilitates end-binding of the MRX complex[17,33].

To ascertain that MR oligomers are active entities, we performed in vitro nuclease assays[21] in the presence of phosphorylated Sae2 (pSae2) to trigger MR endonuclease activity (Fig. 1d-i). Resection was monitored with a radio-labeled oligonucleotide annealing to the produced 3′-overhang. Efficient DNA end resection was observed at the MR concentrations that led to the binding of multiple MR entities on DNA in EMSA assays (Fig. 1c-i and Supplementary Fig. 1g). Interestingly, negative staining TEM analysis of the MR:DNA ratio with the highest resection activity in the nuclease assay (200 nM MR with 2.5 nM DNA, MR:DNA ratio 80:1; Fig. 1d-i) revealed that the MR dimers formed large oligomers on DNA, in which the individual MR dimers could no longer be easily distinguished (Fig. 1d-ii), in contrast to the assemblies with distinguishable MR dimers on DNA observed at lower MR:DNA ratios (Fig. 1b-ii, iii and Supplementary Fig. 1e). The combination of imaging and

biochemical experiments demonstrates that the observed MR oligomers are not inactive assemblies, but rather functional units that allow resection to proceed through multiple endonucleolytic cuts beyond the vicinity of DNA ends, as observed in vivo[6,20].

A high MR:DNA ratio (60:1; Supplementary Fig. 1h) also stabilized MR on DNA against higher salt concentration, thus counteracting the expected salt sensitivity of MR-DNA association[34]. Moreover, MR preferentially bound longer DNA substrates, as shown by the trend of apparent dissociation constants ($K_D$) derived from EMSA experiments with oligonucleotides, which decreased with increasing DNA length (Fig. 1e and Supplementary Fig. 2a). EMSA experiments also revealed that longer DNA molecules stabilized MR on DNA against high salt concentrations (Supplementary Fig. 2b-d), similarly to higher MR:DNA ratios (Supplementary Fig. 1h). As MR-DNA species remaining in the wells may bias the $K_D$ values, we performed mass photometry, in which the large MR-DNA assemblies stay in solution. These measurements confirmed better MR binding to longer oligonucleotides, as the peak area between the two dashed lines, corresponding to one MR dimer binding to one DNA molecule, grew with increasing DNA length (Fig. 1f; theoretical molecular weight of $M_2R_2$ is 460 kDa). Moreover, a higher molecular weight peak at ~1000 kDa could be detected and properly fitted for longer oligonucleotides, corresponding to two MR dimers bound per DNA molecule (zoom-insets for 106 and 197 bp). In addition, larger MR oligomers were likely formed on longer DNA that were beyond the detection limit of the instrument[44], as indicated by the sudden decrease in the observed MR-DNA peak areas with 106 or 197 bp, compared to 73 bp-long DNA (Fig. 1f). Taken together, these observations suggest that MR dimers form oligomers in solution, which dissociate to bind DNA and reassemble in a switch-like manner to larger MR assemblies on DNA (Fig. 1g). As expected, purified MRX behaved comparably in TEM, EMSAs, and mass photometry experiments (Supplementary Fig. 3a–e), demonstrating that Xrs2 binding does not diminish MR oligomerization. MR(X) oligomerization may allow a rapid loading of multiple MR(X) units onto DNA, thereby facilitating short-range resection of DNA ends and foci formation.

**MR oligomerization on DNA accompanies end resection.** Next, we reconstituted short-range (MR and pSae2) and long-range (Exo1) resection in vitro using recombinant *S. cerevisiae* proteins and linear plasmid DNA with free or streptavidin-blocked ends. Gel-based assays confirmed that DNA with free ends could be rapidly processed by Exo1 alone, as visualized by the characteristic pattern of resection products, depending on the extent of resected DNA and whether resection was initiated at one or both DNA ends (Fig. 2a, lane 2, see cartoons on the right; Supplementary Fig. 4a). Adding MR-pSae2 to the Exo1 reaction moderately enhanced resection (Fig. 2a, compare lane 2 with lanes 4, 6 and 8 and Supplementary Fig. 4a), as observed previously[21,45,46]. In contrast, MR-pSae2 alone resulted in slow resection that was limited to the vicinity of DNA ends (Fig. 2a, lanes 3, 5 and 7 and Supplementary Fig. 4a)[6,21]. As expected, long-range resection by Exo1 alone was strongly reduced on linear DNA with streptavidin-blocked ends (Fig. 2b, lane 2 and Supplementary Fig. 4a, b), but this activity could be restored by the addition of MR-pSae2 (Fig. 2b, lanes 4, 6 and 8 and Supplementary Fig. 4a). Thereby, MR-pSae2 was able to overcome the protein block and resect blocked DNA ends similarly to free DNA (Fig. 2b, lanes 3, 5 and 7 and Supplementary Fig. 4a), creating an entry site for Exo1-mediated long-range resection. Using shorter, 3′-labeled oligonucleotide-based DNA with one streptavidin-block at the 5′-terminated end revealed a gradual shortening of the 5′-strand by

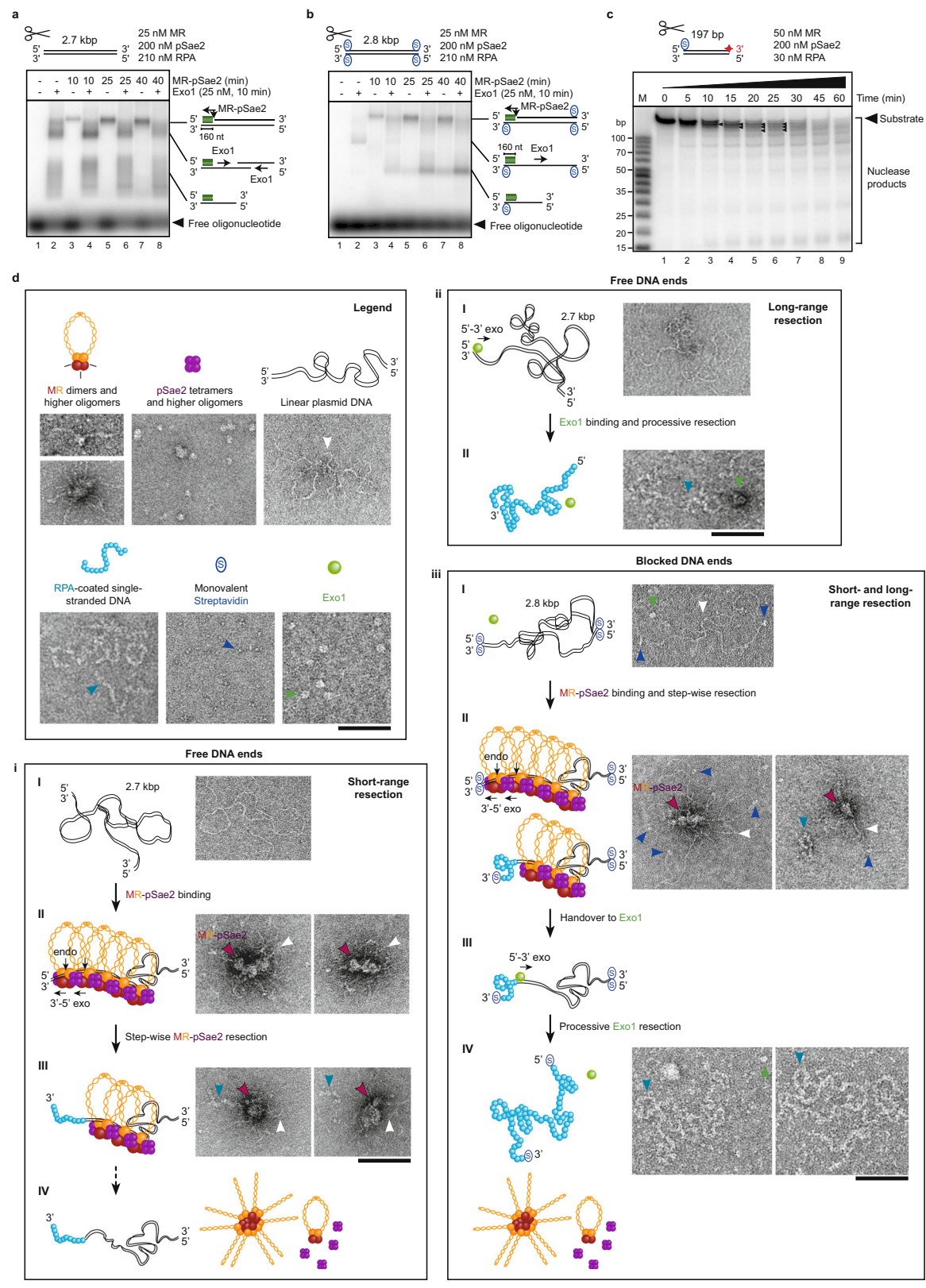

MR-pSae2 and the generation of DNA fragments of similar length (Fig. 2c). This indicates the sequential formation of several endonucleolytic incisions and the stepwise DNA end resection by many MR-pSae2 molecules that together produce the single-stranded overhang observed[21]. MR oligomerization may promote such joint DNA end resection[21]. Together, our reconstituted assays demonstrate that Exo1 requires MR-pSae2-dependent short-range resection at protein-blocked DNA ends to initiate long-range resection.

We next set out to visualize the architecture of DNA end resection complexes, using the same recombinant proteins and reaction conditions, in negative staining TEM. As shown in

**Fig. 2 MR oligomerization on DNA accompanies end resection. a** Short-range end resection of MR-pSae2 and handover to Exo1 for long-range resection (10 min, +) on plasmid DNA with free ends. Resection detection with probe (green box see cartoons) binding to the 3'-overhang produced. Image is representative of $n = 4$ experiments. **b** As **a**, but plasmid DNA with streptavidin-blocked ends (Supplementary Fig. 4a, b). $n = 4$. **c** Stepwise resection by MR-pSae2 (small black arrows) on the 5'-terminated DNA strand. Red star marks the position of the radio-label. $n = 2$. **d** Representative TEM images visualizing short- and long-range end resection by MR-pSae2 and Exo1 on plasmid DNA with free (i, ii) and streptavidin-blocked ends (iii) under conditions comparable to **a** and **b**, $n = 3$. Scale bars: 100 nm. Legend shows cartoons, indicative arrows, and corresponding micrographs of the reaction components. Schematic cartoons in (i–iii) illustrate adjacent micrographs of the different reactions. Due to the resolution of the negative staining images, the number of molecules or their orientation drawn in the cartoons may not exactly correspond to the assembly in the micrographs. (i, I) Linear plasmid substrate shows no significant stretches of RPA-coated, single-stranded DNA in the absence of MR-pSae2 or Exo1. (II–IV) MR-pSae2 oligomerize on DNA (magenta arrows and labels; Supplementary Fig. 4c) and produce short resected, RPA-coated single-stranded DNA overhangs in 2.5 h, before dissociating from DNA and likely forming their respective oligomers. (ii) Exo1 can resect entire plasmid substrates with free ends in 5 min. (iii) Plasmid DNA with two streptavidin blocks per end can only be resected by Exo1 if pSae2-MR oligomers process these DNA substrates beforehand. pSae2-MR clusters (magenta arrows and labels) can harbor several streptavidin-blocked DNA substrates (II, left) and execute short-range resection on one DNA end within 2.5 h, resulting in short resected, RPA-coated single-stranded DNA overhangs (II, right). Exo1 can then fully resect the DNA substrates within 10 min (III–IV).

Fig. 2d, we readily observed MR oligomers in conjunction with pSae2 bound to DNA with free or streptavidin-blocked ends (Fig. 2d-i, II, III, Fig. 2d-iii, II and Supplementary Fig. 4c; magenta arrows and MR-pSae2 labels). While free ends of linear DNA were not clearly visible and likely buried within the MR-pSae2 clusters (Fig. 2d-i, II), streptavidin-blocked ends of linear DNA could be easily distinguished (Fig. 2d-iii,II, dark blue arrows). MR-pSae2 assemblies could bind DNA at internal sites and harbor several DNA molecules, as apparent from the number of streptavidin-blocked ends in particular protein-DNA complexes (Fig. 2d-iii, II, left panel, dark blue arrows), reminiscent of the assembly of repair foci involving several broken DNA molecules in vivo[25,47]. Interestingly, resection of DNA ends was observed in TEM independently of protein blocks, producing up to ~300 nt-long stretches of single-stranded DNA, consistent with cellular[6,20] and previous biochemical data[21]. These single-stranded tracts were readily coated by RPA, thus appearing more electron-dense than double-stranded DNA (Fig. 2d-i, III, Fig. 2d-iii, II, right panel; arrows in light blue). In accordance with the biochemical assay in Fig. 2a, MR-pSae2 activity was not required to trigger Exo1-dependent long-range resection on DNA templates with free ends, as visualized by the appearance of plasmid-length RPA-coated, single-stranded DNA upon incubation with Exo1 alone (Fig. 2d-ii, II, arrows in light blue). In contrast, MR-pSae2-dependent end processing was required for subsequent tail elongation by Exo1 when using DNA substrates with blocked ends (Fig. 2d-iii, II–IV). Accordingly, no RPA-coated, single-stranded DNA overhangs were observed when imaging streptavidin-blocked DNA with Exo1 without prior short-range resection by MR-pSae2 to remove the protein block (Fig. 2d-iii, I). Taken together, these single-molecule assays recapitulated that short-range resection by MR-pSae2 is essential for initial processing of protein-blocked ends to allow handover to the Exo1 exonuclease for subsequent long-range resection. Strikingly, our TEM analysis visualized large MR-pSae2 assemblies at DNA ends undergoing resection.

**Mutagenesis of conserved Rad50 interface regulates MR oligomerization**. To define the functional relevance of MR oligomerization on DNA, we next set out to identify the interaction interface(s) between the MR head domains. As a basis for mutational analysis, we employed a recent crystallographic structure showing a eukaryotic *C. thermophilum* (*Ct*) Rad50 dimer bound to a DNA molecule that forms a "quasi-continuous" structure with another Rad50 dimer-bound DNA molecule adjacent in the crystal, indicating a possible arrangement of two Rad50 dimers on DNA (Fig. 3a; RCSB Protein Data Bank PDB identifier: 5DAC[48], https://doi.org/10.2210/pdb5dac/pdb). We hypothesized that an

anti-parallel, two-stranded beta-sheet (black rectangle, β) and its connecting loop (black rectangle, L) protruding from the head domain of one Rad50 dimer in one asymmetric unit towards the identical structure of another Rad50 dimer in the adjacent asymmetric unit of the crystal could represent a possible interaction site, especially on a continuous DNA molecule and in solution when MR molecules could come even closer on DNA than in the crystal structure shown. The residues constituting this small beta-sheet and its connecting loop, or their physicochemical characteristics, are conserved between *C. thermophilum*, *S. cerevisiae*, and *H. sapiens* Rad50 (Fig. 3b), which is intriguing given the limited overall sequence similarity of MR between species[49]. Interestingly, mutations derived from cancer predisposed individuals (* in Fig. 3b, from ClinVar/MedGen databases) cluster in this conserved beta-sheet and could be relevant for Rad50 functions in DNA repair and cancer avoidance.

To investigate whether beta-sheet interactions are indeed involved in MR oligomerization, we stained *S. cerevisiae* (*Sc*) MR oligomers with Thioflavin T (ThT), a dye used to detect beta-sheet-rich structures[50]. Indeed, a ThT signal was detected in the presence of *Sc*MR oligomers at 4 °C and enhanced at higher temperature (42 °C), suggesting the presence of extended beta-sheet interactions between MR molecules (Fig. 3c). In order to test the possible role of the identified protruding beta-sheet and its connecting loop in MR oligomerization, we engineered two *Sc*Rad50 variants, each with several point mutations in the beta-sheet to disrupt either electrostatic or hydrophobic interactions between the Rad50 subunits of MR dimers. Addressing the electrostatic component of MR oligomerization (Supplementary Fig. 1c), we substituted a polar residue in the potential interaction interface with an uncharged, non-polar alanine (*Sc* N121A, *Ct* N122) and removed a negative charge (*Sc* D124, *Ct* E125) to disrupt an attractive salt bridge between the D124 and R125 residues (*Ct* E125, R126) or to alleviate a potentially strong repulsive force between the aspartate residues D124 (*Ct* E125) of interacting *Sc*Rad50 molecules (red residues mutated in Fig. 3a zoom-inset and Fig. 3b; Rad50^ho). In the second *Sc*Rad50 variant, four hydrophobic beta-sheet residues were mutated to alanine to disrupt or destabilize the beta-sheet structure[51] (*Sc*Rad50^L116A/I119A/T127A/L128A; *Ct*Rad50^L118/L120/T127/I129; blue residues mutated in Fig. 3a zoom-inset and Fig. 3b; Rad50^lo). These two *Sc*Rad50 mutants were co-expressed and co-purified with Mre11, showing that the physical interaction between Mre11 and Rad50 was not notably affected (Supplementary Fig. 5a).

We first characterized the oligomerization capacity of the *Sc*MR variants in negative staining TEM in the absence of DNA. This analysis revealed that the quadruple mutant MR^L116A/I119A/T127A/L128A formed only dimers (thus

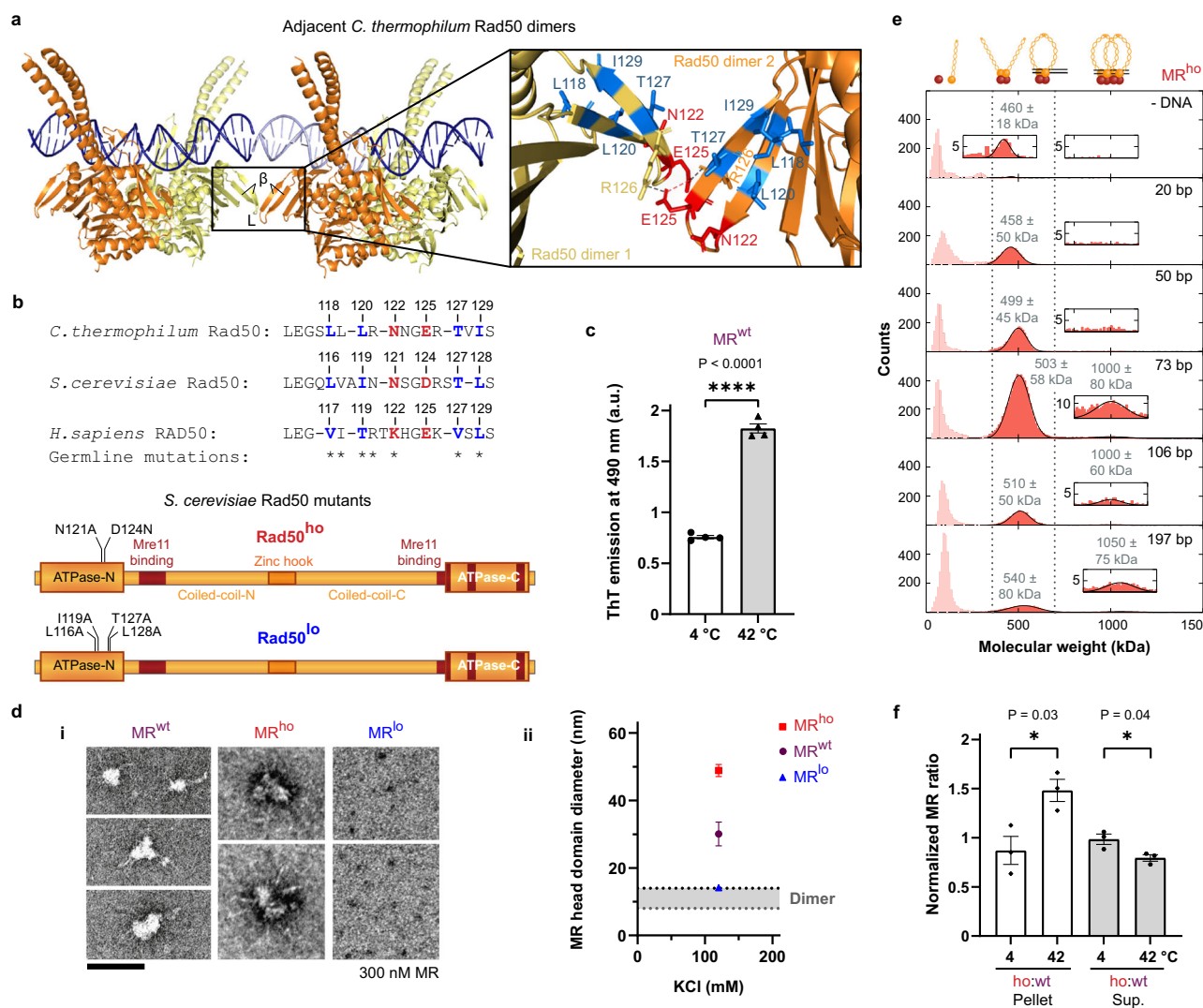

**Fig. 3 Mutagenesis of conserved Rad50 interface regulates MR oligomerization. a** Crystal structure of two *C. thermophilum* (*Ct*) Rad50 dimers bound to one oligonucleotide molecule each (PDB: 5DAC[48], 10.2210/pdb5dac/pdb) used to generate a plausible interaction model as guide for validation by mutagenesis. Possible interactions of adjacent Rad50 dimers on DNA *via* a small beta-sheet (β) and its connecting loop (L) protruding from the Rad50 head domain (rectangle), especially if bound DNA was continuous. Zoom-inset shows red/blue residues mutated in the *S. cerevisiae* (*Sc*) Mre11-Rad50[ho/lo] variants. Gray dotted line: salt bridge between E125 and R126 residues of adjacent *Ct*Rad50 dimers (spacing 3.8 Å within crystal). Red dotted line: repulsive E125 residues of adjacent *Ct*Rad50 dimers are spaced by 4.8 Å within the crystal, but could come closer on continuous DNA and in solution. **b** Sequence alignment of the *Ct*, *Sc*, and *H. sapiens* Rad50 beta-sheet motifs. Color code as in **a**. * indicates germline point mutations listed on ClinVar/MedGen for cancer-predisposed humans. Cartoons show multiple point mutations introduced in *Sc*Rad50 to disrupt electrostatic (Rad50[ho]) or hydrophobic (Rad50[lo]) interactions (Supplementary Fig. 5a). **c** Thioflavin T (ThT) fluorescence of *Sc*MR[wt] indicates beta-sheet rich higher-order structures (Supplementary Fig. 5e). Mean ± SEM, n = 4 independent experiments, unpaired two-tailed *t*-test, a.u., arbitrary units. **d** (i) Representative TEM of *Sc*MR[wt/ho/lo] as in Fig. 1a–v. (ii) Quantification of mean head domain diameters ± SEM of *Sc*MR[wt/ho/lo] oligomers from (i) along their longest axis. $n_{MR\text{-}wt} = 30$ molecules, $n_{MR\text{-}ho} = 72$, and $n_{MR\text{-}lo} = 46$. Gray area indicates diameter range of ~8–14 nm measured for *Sc*MR[wt] dimers. Scale bar: 100 nm. **e** Molecular mass distributions from mass photometry show better *Sc*MR[ho] binding to longer oligonucleotides (100 nM base pairs). Cartoons illustrate species in peaks. Measured molecular weight (MW, kDa) ± SD in gray. Zoom-insets show MW range where a peak of two MR dimers bound to one DNA molecule would be expected and was fitted where possible (Supplementary Fig. 3b). n = 3 independent experiments. **f** Quantification of *Sc*MR[wt/ho] pelleting assays at 4 or 42 °C (see Supplementary Fig. 5c). Sup. supernatant. Higher temperature promotes oligomerization. Mean ± SEM, n = 3, unpaired two-tailed *t*-tests.

henceforth denoted low oligomerization mutant, MR[lo]), while the MR[N121A/D124N] mutant (henceforth denoted high oligomerization mutant, MR[ho]) assembled to even larger oligomers than wild-type MR (MR[wt]) (Fig. 3d). Mass photometry confirmed that the MR[ho] mutant formed larger oligomers than MR[wt], as in the absence of DNA only a small dimeric fraction of MR[ho] could be detected (−DNA, zoom-inset ~460 kDa; Fig. 3e and Supplementary Fig. 5b). The MR[ho] mutant was also more resistant towards high salt (240 mM KCl), as significantly

more MR[ho] oligomer assemblies could be harvested in a pelleting assay compared to MR[wt] after incubation at 1.2 µM MR and higher temperature (42 °C) (42 °C pellet; Fig. 3f and Supplementary Fig. 5c). Consistently, mass photometry detected less MR[ho] dimers at ~460 kDa after 30 min incubation at 250 mM KCl and 30 °C compared to MR[wt], suggesting the formation of larger MR[ho] assemblies beyond the detection limit (Supplementary Fig. 5d). Similar to MR[wt], MR[ho] was stabilized on longer oligonucleotide substrates (Figs. 1f and 3e). However,

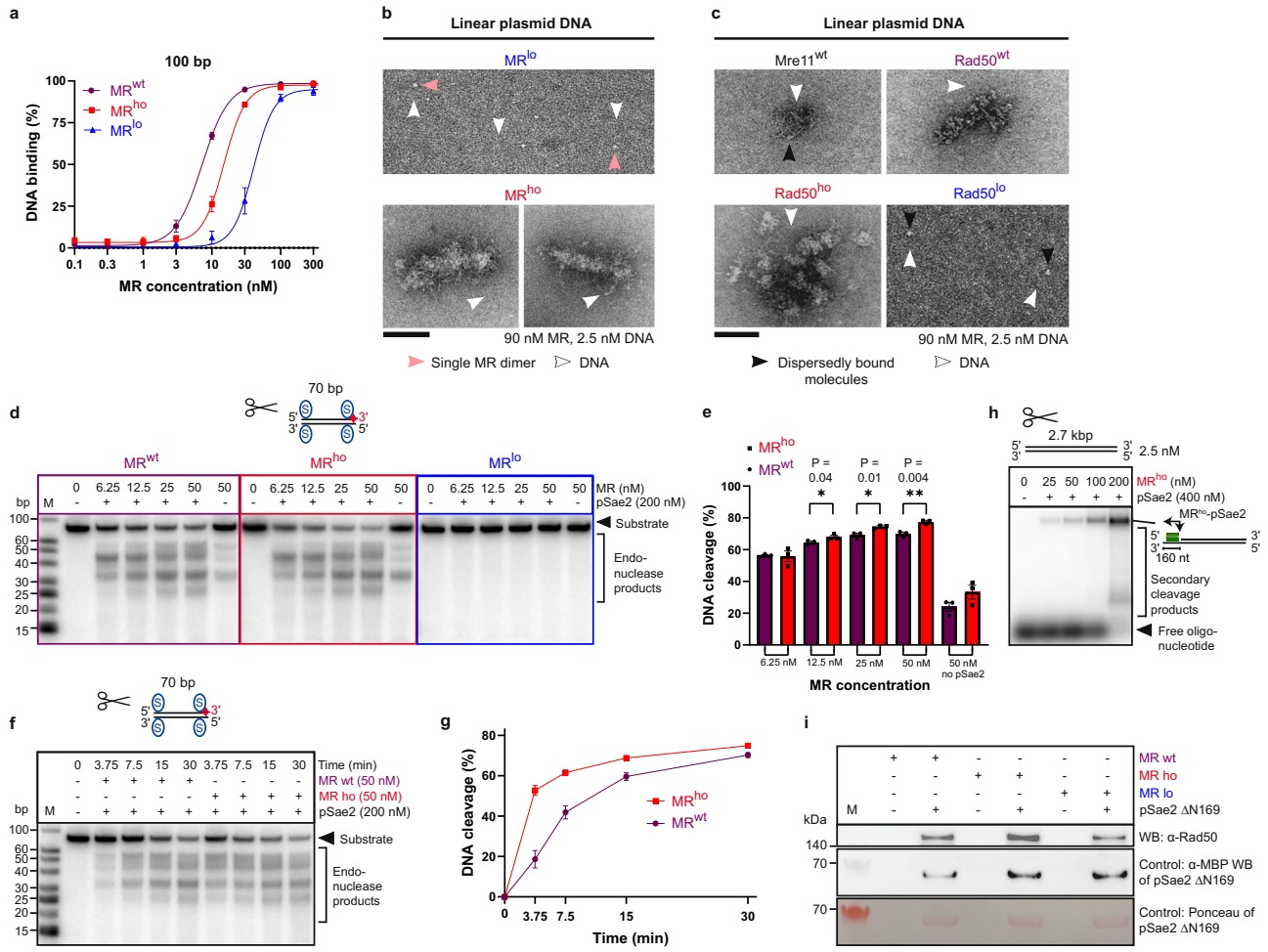

**Fig. 4 MR oligomerization influences DNA binding behavior and endonuclease activity. a** Quantification of EMSAs of MR$^{wt/ho/lo}$ binding to 100 bp-long DNA as in Supplementary Figs. 2a and 5f. Mean ± SEM, $n = 3$ independent experiments, sigmoidal fit. **b** TEM of MR$^{ho/lo}$ binding to plasmid DNA. Images representative of $n = 3$ experiments. MR$^{lo}$ binds dispersedly on DNA without oligomerization, while MR$^{ho}$ dimers oligomerize on DNA to a denser "worm"-like structure compared to the "pearls-on-a-string" assembly of MR$^{wt}$ (Fig. 1b-ii) at similar conditions (see "Methods"). **c** TEM of Mre11$^{wt}$ and Rad50$^{wt/ho/lo}$ binding to plasmid DNA. $n = 3$. In contrast to purified Mre11$^{wt}$, Rad50$^{wt}$ forms "pearls-on-a-string"-like structures on linear plasmid DNA. Rad50$^{lo}$ fails to oligomerize, thus binds dispersedly on DNA, while Rad50$^{ho}$ clusters into large oligomers on DNA, similarly to MR$^{ho}$. **d, e** Endonuclease assays of MR$^{wt/ho/lo}$-pSae2 (**d**, Supplementary Fig. 6a) with quantification (**e**). Images representative of $n = 3$ experiments. Mean ± SEM, unpaired two-tailed $t$-tests. Red star marks the position of the radio-label. **f, g** Representative endonuclease activity kinetics of MR$^{wt/ho}$-pSae2 (**f**) with quantification (**g**). Mean ± SEM, $n = 3$ independent experiments. Red star marks the position of the radio-label. **h** Representative resection assay of MR$^{ho}$ with probe (green box in cartoon) binding to the 3'-overhang produced. $n = 2$. Note that the large MR$^{ho}$ oligomers in Fig. 4b are active in resection. **i** Interaction assay of MR$^{wt/ho/lo}$ with the C-terminal part of pSae2 (pSae2 ΔN169). Ponceau and anti-MBP western blot indicate elution of MBP-tagged pSae2 ΔN169. The anti-Rad50 western blot shows interaction of pSae2 ΔN169 with Rad50 in MR$^{wt/ho/lo}$. One out of two independent experiments is shown. Scale bars: 100 nm (**b**, **c**).

MR$^{ho}$ formed larger oligomers on longer oligonucleotide DNA compared to MR$^{wt}$, as the peak at ~1000 kDa of two MR$^{ho}$ dimers binding to one DNA molecule could be detected and properly fitted already for 73 bp with a larger peak area than for MR$^{wt}$ (compare 73 bp zoom-insets in Figs. 1f and 3e). Moreover, the areas of the MR-DNA peaks decreased more with 106 and 197 bp DNA for MR$^{ho}$ than MR$^{wt}$, indicating the formation of larger MR$^{ho}$ nucleoprotein assemblies beyond the detection limit (compare peak areas for 106 and 197 bp in Figs. 1f and 3e). For MR$^{lo}$, on the other hand, binding to oligonucleotide DNA could not be observed in mass photometry at the low MR and DNA concentrations required for measuring (197 bp in Supplementary Fig. 5b).

**MR oligomerization influences DNA binding behavior and endonuclease activity.** EMSA experiments revealed a moderate

DNA binding impairment of the MR$^{lo}$ mutant compared to MR$^{wt}$ (Fig. 4a and Supplementary Figs. 2a and 5f); however, at ~90 nM, MR$^{lo}$ efficiently bound the 100 bp-long DNA substrate. Similarly, 90 nM MR$^{lo}$ was also bound to linear plasmid-length DNA (35 mM KCl, MR$^{lo}$:DNA 36:1) in negative staining TEM, appearing as dispersed single dimers (Fig. 4b). Surprisingly, the DNA binding of the MR$^{ho}$ mutant was also reduced in EMSA experiments (Fig. 4a and Supplementary Figs. 2a and 5f), indicating that higher oligomeric structures may interfere with disassembly to smaller MR$^{ho}$ complexes including dimers and thus efficient loading onto DNA. Mass photometry measurements at low salt (25 mM KCl, zoom-insets in Supplementary Fig. 5g) revealed that once MR$^{ho}$ loaded onto plasmid DNA, it assembled as one or maximally two MR$^{ho}$ dimers per DNA molecule (in 1500–2500 kDa range, zoom-inset in Supplementary Fig. 5g) and then rapidly formed large nucleoprotein assemblies beyond the detection limit. Thus, the tail of events >2500 kDa, corresponding

to smaller nucleoprotein assemblies, was significantly reduced for MR$^{ho}$ compared to MR$^{wt}$ (25 mM KCl, zoom-insets in Supplementary Fig. 5g), and even more so after incubation (25 mM KCl, 30 min, zoom-insets in Supplementary Fig. 5g). Moreover, TEM analysis demonstrated that MR$^{ho}$ assembled into defined, condensed "worm"-like structures once it bound plasmid DNA (90 nM MR$^{ho}$, 35 mM KCl, MR$^{ho}$:DNA 36:1; Fig. 4b), while MR$^{wt}$ formed much smaller oligomers at comparable conditions (100 nM MRwt; Fig. 1b-ii and Supplementary Fig. 1e). We next expressed the Mre11$^{wt}$ and Rad50$^{wt/ho/lo}$ subunits individually and observed that Mre11$^{wt}$ alone did not oligomerize on DNA (Fig. 4c). In contrast, oligomerization of the Rad50$^{wt/ho/lo}$ subunits recapitulated the behavior of the respective MR heterodimers (Figs. 1b-ii and 4b, c): Rad50$^{ho}$ formed larger oligomeric assemblies on DNA compared to Rad50$^{wt}$, while Rad50$^{lo}$ bound DNA dispersedly. Taken together, these results suggest that Rad50 is necessary and sufficient to drive oligomerization of the MR complex, most likely through a conserved beta-sheet motif.

We next performed clipping assays using oligonucleotide-based DNA with blocked ends to assess if MR oligomerization affects its endonuclease activity in conjunction with pSae2. Interestingly, the MR$^{lo}$ mutant was entirely endonuclease-deficient, even when used at a concentration (200 nM) that supports DNA binding (Fig. 4d, e and Supplementary Fig. 6a). Conversely, MR$^{ho}$ was slightly more active than MR$^{wt}$ (Fig. 4d, e), which was particularly apparent upon short incubation times during kinetic experiments with oligonucleotide-based DNA (Fig. 4f, g). Using plasmid-length substrates, we observed that conditions resulting in large "worm"-like assemblies support DNA end resection (90–100 nM MR$^{ho}$, Fig. 4b, h). A further increase of the MR$^{ho}$ concentration to 200 nM additionally enhanced resection activity (Fig. 4h), suggesting that the assemblies are sites of efficient DNA end resection. The initial resection by MR$^{ho}$ could efficiently be extended by Exo1 (Supplementary Fig. 6b–d), similarly to MR$^{wt}$ (Fig. 2a, b), indicating that the large protein assembly does not interfere with downstream long-range resection.

Notably, both MR$^{lo}$ and MR$^{ho}$ were proficient as exonucleases and functionally interacted with ATP (Supplementary Fig. 6e). Moreover, both MR variants were able to bind pSae2 (Fig. 4i), as shown in a protein interaction assay using the conserved C-terminal part of Sae2 (pSae2 △N169)[19]. These experiments demonstrate that although endonuclease-deficient, the Rad50$^{lo}$ variant is clearly distinct from the Rad50S mutants, which fail to physically and functionally interact with Sae2[19,52]. Taken together, these data suggest that the conserved beta-sheet motif in Rad50 regulates MR oligomerization, and that oligomerization in turn promotes MR resection activity.

**Foci formation and DNA repair require MRX oligomerization in vivo**. In order to probe the function of MRX oligomerization in vivo, we introduced Rad50$^{wt}$ or the Rad50$^{ho/lo}$ mutants in an *S. cerevisiae* Rad50 deletion (*rad50△*) strain together with a C-terminal yEVenus- tag (*rad50$^{wt}$-yEVenus, rad50$^{ho}$: rad50$^{N121A/D124N}$-yEVenus, rad50$^{lo}$: rad50$^{L116A/I119A/T127A/L128A}$-yEVenus*), while Mre11 and Xrs2 were unaltered. DNA damage sensitivity assays with the topoisomerase inhibitor and DSB-inducer camptothecin (CPT), as well as the replication inhibitor hydroxyurea (HU), revealed that cells expressing Rad50$^{ho}$ coped with DNA lesions similarly to wild-type, while cells with Rad50$^{lo}$ displayed a phenotype comparable to *rad50△* (Fig. 5a). This defect was recapitulated by proliferation assays with 5 or 25 µM CPT: whereas the doubling times of wild-type and Rad50$^{ho}$-expressing cells only marginally increased upon CPT-treatment, cells with Rad50$^{lo}$ or *rad50△* were more sensitive to DNA damage, especially at higher CPT concentrations (Fig. 5b). We then engineered wild-type yeast strains overexpressing untagged Rad50$^{wt/ho/lo}$ from the galactose-inducible *GAL1*-promoter (Fig. 5c, d). We found that overexpression of Rad50$^{lo}$ on top of endogenous Rad50$^{wt}$ results in a pronounced growth defect and DNA damage sensitivity (Fig. 5d). Although we cannot exclude that the introduced mutations affect the endonuclease activity by altering the local structure, these results suggest that Rad50$^{lo}$ retains the capacity to act in a dominant-negative manner, most likely by inhibiting oligomerization of the wild-type MRX pool.

Live-cell imaging of yEVenus-tagged Rad50 variants revealed the formation of small Rad50$^{wt/ho}$ foci at potential DNA damage sites in the nucleus of CPT-treated cells (arrows, Fig. 5e). Interestingly, the Rad50$^{lo}$ mutant instead failed to assemble distinguishable foci (Fig. 5e), although we cannot exclude that its slightly reduced protein levels contribute to this defect (Fig. 5f). Conversely, Rad50$^{ho}$ formed foci even in the absence of CPT-induced DNA damage (Fig. 5e, g and Supplementary Fig. 7a). These data support the notion that MRX oligomerization promotes stable DNA binding required for the assembly of large repair structures detectable as foci in vivo.

**DNA damage checkpoint activation and telomere maintenance**. Upon DNA damage, the checkpoint-kinase Tel1 is recruited by MRX to repair foci at DNA lesions and phosphorylates target proteins including Rad53, which delays cell cycle progression until the DNA damage is repaired[5]. A single unrepaired DSB is sufficient to trigger the DNA damage checkpoint in yeast[53]. We hypothesized that MRX oligomerization at DNA lesions could have a function in amplifying the checkpoint signal. Indeed, the phosphorylation-mediated mobility shift of Rad53 upon DNA damage was reduced in cells expressing untagged Rad50$^{lo}$ compared to Rad50$^{wt}$ or Rad50$^{ho}$ in western blotting; however, the limited Rad53 phosphorylation was still sufficient to allow full arrest of Rad50$^{lo}$-expressing cells after HU treatment, as demonstrated by flow cytometry (Fig. 6a and Supplementary Fig. 7b). MRX and Tel1 are, however, also required for the maintenance of telomere length — a function that does not require the nuclease activity of the MRX complex[54]. A characteristic feature of *rad50△* cells are thus shortened telomeres (Fig. 6b). Telomeric PCR experiments revealed that yeast cells carrying untagged Rad50$^{lo}$ suffered from short telomeres similarly to *rad50△*, while cells with Rad50$^{ho}$ maintained normal telomere length (Fig. 6b, c and Supplementary Fig. 7c). Taken together, these data suggest that MRX oligomerization at DNA damage sites, governed by the conserved Rad50 beta-sheet motif, facilitates the assembly of protein foci, which is required for the DNA repair, signaling, and telomere maintenance functions of MRX in vivo.

## Discussion

Here, we show that *S. cerevisiae* MR(X) dimers oligomerize *via* the head domain of Rad50 to nucleolytically active assemblies at DNA ends. The formation of these higher-order MR complexes is governed by a conserved beta-sheet motif in the Rad50 head domain. Mutational analysis of the beta-sheet motif identified specific mutations that preserve MR dimerization but decrease or enhance MR head domain oligomerization in vitro. Biochemical reconstitutions coupled with TEM imaging revealed that MR oligomerization is required for pSae2-stimulated endonucleolytic DNA cleavage and hence short-range resection but is dispensable for MR exonuclease activity. Subsequent analysis of the Rad50 mutants in vivo confirmed the importance of MRX oligomerization for its cellular functions in DNA repair, signaling, and telomere maintenance.

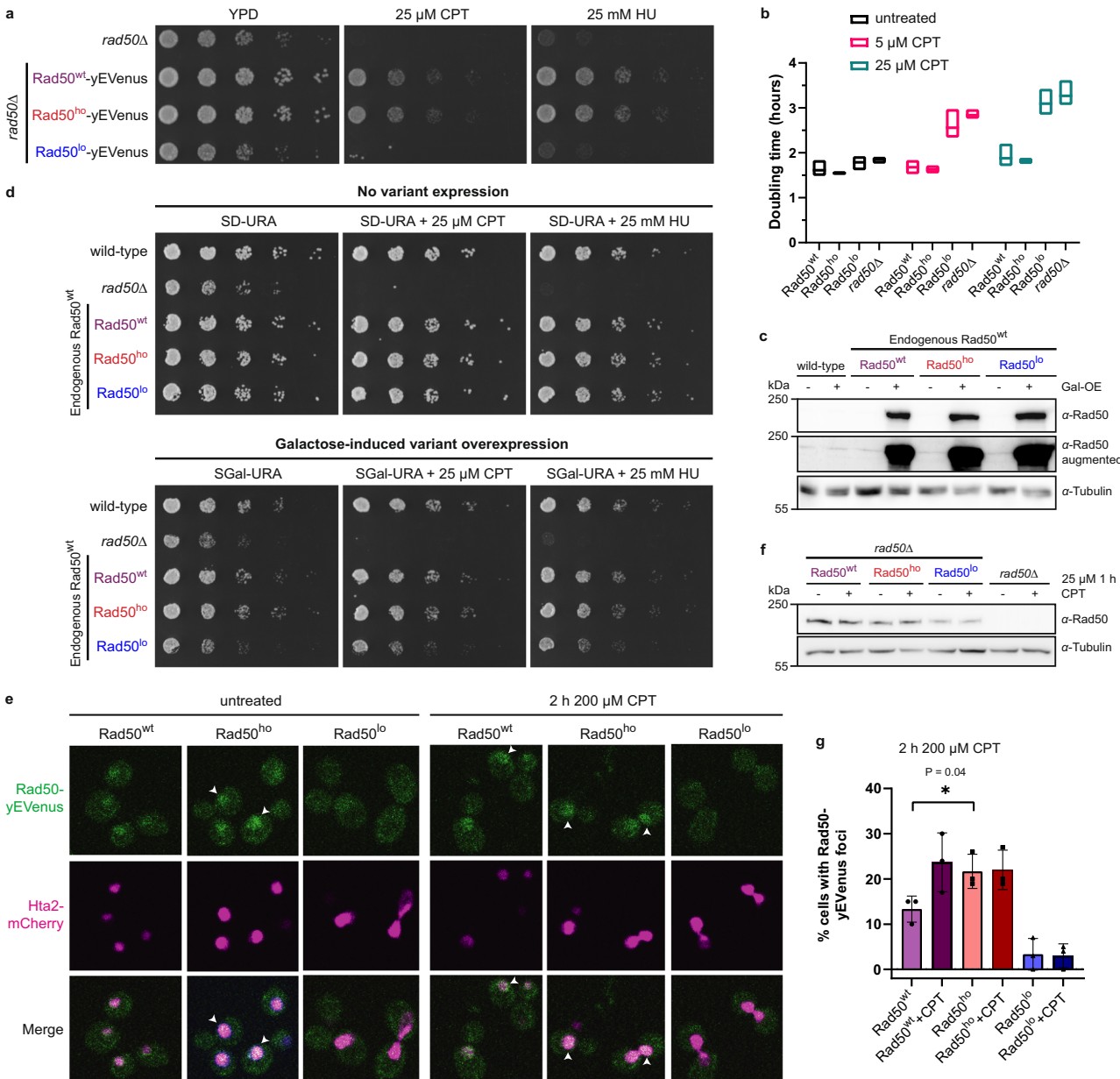

**Fig. 5 Foci formation and DNA repair require MRX oligomerization in vivo. a** Serial dilution of exponentially growing *S. cerevisiae rad50*Δ cells expressing Rad50^wt/ho/lo-yEVenus, spotted on agar plates (YPD) +/− DNA damage-inducing camptothecin (CPT) or hydroxyurea (HU), and incubated for 2 days at 30 °C. *n* = 3. **b** Doubling times of exponentially growing *rad50*Δ cells and *rad50*Δ strains expressing Rad50^wt/ho/lo-yEVenus +/− CPT treatment. Mean and minimum to maximum values shown, *n* = 3 independent experiments. **c** Anti-Rad50 western blot showing galactose-induced overexpression of untagged Rad50^wt/ho/lo in wild-type cells (+Gal-OE). Center panel is contrast-enhanced to show endogenous Rad50^wt levels (wild-type, −Gal-OE). α-Tubulin controls equal loading. *n* = 3. **d** Serial dilution of exponentially growing wild-type cells expressing galactose-inducible Rad50^wt/ho/lo, spotted on SD-URA agar plates with glucose (−) or galactose (+ variant overexpression), and incubated at 30 °C for 2 days. Before overexpression, cells carrying Rad50^wt/ho/lo were similarly resistant to DNA damage as wild-type cells due to endogenous Rad50^wt (−, top). *n* = 3. **e** Representative live-cell images of untreated or CPT-treated *rad50*Δ cells expressing Rad50^wt/ho/lo-yEVenus with histone Hta2-mCherry as nuclear marker. Arrows indicate nuclear Rad50 foci (Supplementary Fig. 7a). **f** Anti-Rad50 western blot showing expression levels of Rad50^wt/ho/lo-yEVenus in *rad50*Δ cells before (−) and after (+) CPT treatment. α-Tubulin controls equal loading. *n* = 3. **g** Quantification of live-cell imaging in **e**. The percentage of cells with nuclear Rad50-yEVenus foci (arrows in **e**) is shown for untreated and CPT-treated *rad50*Δ cells expressing Rad50^wt/ho/lo-yEVenus. Mean ± SD, *n* = 3 independent experiments, unpaired two-tailed *t*-test. Scale bars: 5 µm.

Using reconstituted biochemical reactions, we could visualize intermediates of DNA end resection in vitro. Our biochemical assays coupled with TEM analysis recapitulated MR-pSae2-dependent short-range processing and subsequent tail elongation by Exo1. TEM imaging revealed that oligomerization of MR and pSae2 complexes to higher-order assemblies can occur

internally of DNA, but resection is restricted to DNA ends. By sliding along DNA as previously shown[55], MR could find and accumulate at the DNA end, where the MR endonuclease is locally activated by pSae2. The resulting short-range processing of blocked ends is prerequisite for subsequent handover to Exo1-mediated long-range resection. After cleavage, MR-pSae2

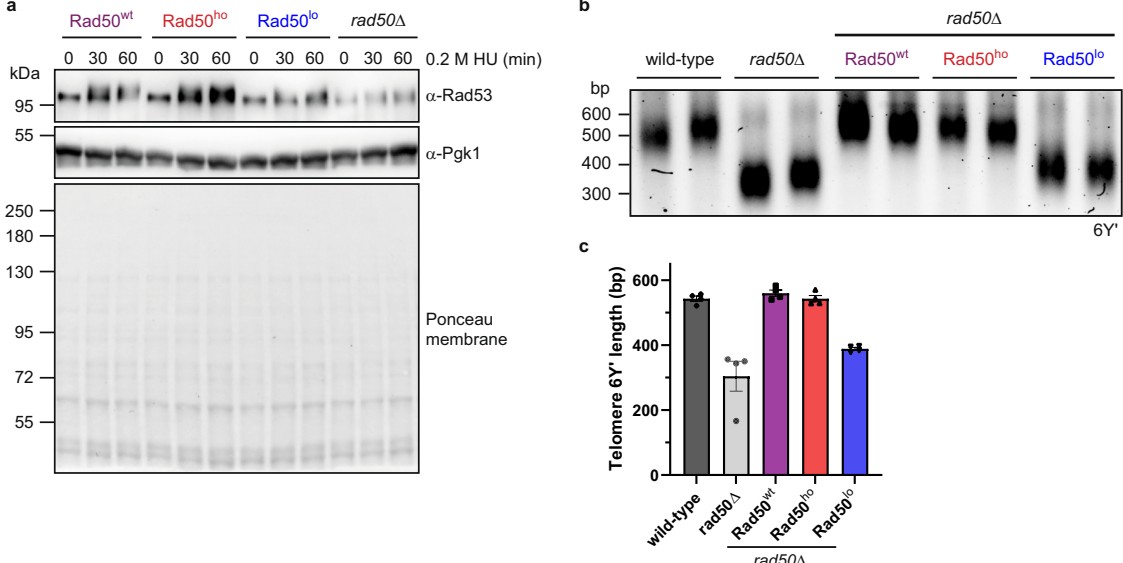

**Fig. 6 DNA damage checkpoint activation and telomere maintenance. a** Activation of the Rad50^wt/ho/lo-mediated DNA damage checkpoint, monitored by HU-induced Rad53 phosphorylation (α-Rad53). In contrast to untagged Rad50^wt/ho, the upshift of the phosphorylated Rad53 band is reduced in *rad50Δ* and Rad50^lo-expressing cells (see also Supplementary Fig. 7b). α-Pgk1 controls equal loading. *n* = 2. **b** Representative telomeric PCR analysis for telomere 6Y′ in two clones per genotype. PCRs were performed with oligonucleotides oBL361 and oBL359 (see also "Methods" and Supplementary Fig. 7c). *n* = 4. **c** Quantification of telomeric PCR analysis of telomere 6Y′ in **b**. Fragment length was determined by using the 100 bp DNA ladder as size reference. *n* = 4 for each genotype; mean ± SD. Note that *rad50Δ* and Rad50^lo-expressing cells have shorter telomeres than cells with Rad50^wt/ho.

assemblies dissociate from RPA-coated single-stranded DNA ends to allow subsequent steps in the DSB repair pathway.

Previous work showed that an initial endonucleolytic cut by MRX-pSae2 is positioned 15–35 nt away from the DNA end when using oligonucleotide-based DNA[12]. On longer DNA templates, however, additional MRX-pSae2 incisions were found that led to the resection of several hundred nucleotides in length[21], consistent with observations from meiotic cells lacking the long-range resection pathways[6,20]. Accordingly, a stepwise DNA cleavage model for 5′-DNA end resection by MRX-Sae2 was proposed[21]. Here, our experiments revealed that MR molecules form large assemblies, which provide a structural basis to explain the multiple endonucleolytic cuts along the 5′-terminated DNA strand during short-range resection.

Mutations in a protruding beta-sheet in the Rad50 subunit allowed us to construct two MR mutants, one with enhanced (MR^ho) and one with reduced oligomerization (MR^lo). Although available results demonstrate that MR oligomerization is dependent on the Rad50 head domain and beta-sheet interactions, the resolution of the TEM images does not allow to prove that the beta-sheet motif directly mediates this effect. Nevertheless, MR^ho shows slightly increased resection activity compared to MR^wt, and enhanced foci formation in living cells. As MR^ho displays enhanced oligomerization despite the loss of an attractive salt bridge between *Sc* D124 and R125 (*Ct* E125 and R126) due to the D124N mutation, we propose that the abolished charge repulsion of D124N residues between MR^ho molecules could drive this behavior. Although in the crystal structure model with two separate and only "quasi-continuous" DNA molecules (Fig. 3a) the *Ct* E125 (*Sc* D124) residues of adjacent Rad50 dimers are ~5 Å apart, they may be closer on a continuous DNA molecule in solution. Moreover, the hydrophobic residues in the beta-sheet could come closer when charge interactions are alleviated, which could explain the lower salt-sensitivity of MR^ho oligomers and the more pronounced beta-sheet signal in ThT staining. Yet, other head domain interactions are possible and high-resolution structures of MR dimers adjacently bound on DNA will

ultimately be needed to demonstrate the exact residues mediating beta-sheet-driven MR oligomerization.

In contrast to MR^wt and MR^ho, MR^lo was endonuclease-deficient in vitro and failed to form DNA damage-induced nuclear foci in vivo, resulting in cellular sensitivity to DNA damage, an altered checkpoint response and short telomeres. Moreover, Rad50^lo acts as a dominant-negative factor on endogenous Rad50^wt in overexpression experiments in yeast cells. Although we cannot exclude that the mutations in MR^lo negatively affect MR^lo activity beyond disrupting oligomerization, we note that oligomerization accompanies resection in MR^wt, and that the Rad50^lo mutant still binds pSae2, Mre11, and Xrs2 that mediates the observed nuclear localization in cells[17]. Moreover, MR^lo shows normal levels of exonuclease activity, which is inhibited by ATPγS, as observed for wild-type MR[21], indicating that the nuclease active site of MR^lo is sufficiently intact, given that the endo- and exonuclease activities of MR are mediated through the same active site[56]. The defects of MR^lo are also unlikely explained with reduced DNA binding, as increased MR^lo concentrations restored DNA binding but not its enzymatic activity. In addition, MR^ho was more active as endonuclease than MR^wt, despite being impaired in DNA binding as well. We thus propose that MR oligomerization on DNA might facilitate the drastic conformational changes required for nuclease activation[38,57], and promote activation of adjacent MR molecules on the 5′-terminated DNA strand, thereby creating a "domino-like" effect close to the DNA ends that explains step-wise short-range resection.

Repair foci with hundreds to thousands of MRX/N molecules were found at DNA breaks in yeast and mammalian cells[23–28]. ChIP-seq data of human MRN show an accumulation of the complex across a DSB on both sides of broken DNA, covering ~1 kb of DNA, with the strongest MRN localization at the DNA ends, indicating MRN focus formation around a DSB and tethering of broken ends[58]. In accord, our TEM analysis revealed that MR clusters can hold several DNA molecules together, mediated by oligomerization of the MR head domains and

interactions *via* the zinc hooks. We thus propose that MR head domain oligomerization together with DNA end tethering may explain the coordinated resection observed at two-ended DNA breaks[59]. Consistent with this interpretation, no nuclear foci were detected upon DNA damage for yEVenus-tagged oligomerization-defective MR[lo]X in *S. cerevisiae*, while the hyper-oligomerizing MR[ho]X mutant formed nuclear foci even in the absence of exogenous DNA lesions. Moreover, pelleting and ThT assays detected stable MR assemblies, which could build a rather rigid core inside potentially liquid-like repair foci, as formed, e.g., by the NHEJ-factor 53BP1[60] or the strand-exchange protein Rad52[61], and act as scaffold for transiently joining downstream repair and signaling components including pSae2/pCtIP and Tel1/ATM.

Interestingly, database (ClinVar/MedGen) and literature reports[62,63] suggest that human individuals with a hereditary predisposition for BRCA1/2- and PALB2-associated cancers carry germline point mutations of predominantly hydrophobic residues in the human RAD50 beta-sheet motif. The tumor suppressor BRCA1 interacts with PALB2 and BRCA2, and localizes to repair foci at DNA lesions similarly to MR[64–66]. Loss of BRCA1/2 function leads to aberrant degradation of nascent DNA by MRE11 upon replication stress, and a dramatic increase in the lifetime risk for ovarian and breast cancer[3,65–67]. The abrogation of MRE11 activity in BRCA1/2-deficient cells, on the other hand, was linked to PARP inhibitor resistance in cancer chemotherapy[2]. It is intriguing that MR[lo], bearing beta-sheet-disturbing alanine mutations of four hydrophobic residues altered in cancer pre-disposed individuals, shows impaired endonuclease activity in vitro. These loss-of-function RAD50 mutations could restrict MRE11-mediated degradation of nascent DNA at challenged replication forks in PARP-resistant, BRCA1/2-PALB2-mutated cells, and may help to explain tumor chemoresistance. However, the importance of these RAD50 mutations for cancer risk and survival is not well understood and thus more extensive studies are needed.

## Methods

**Preparation of recombinant proteins**. Yeast Mre11-Rad50 complexes were co-expressed in *S. frugiperda* 9 (*Sf*9) cells with an optimal ratio of baculoviruses producing Mre11[wt]-6xHis and untagged Rad50[wt], Rad50[ho], or Rad50[lo] that was codon-optimized for *Sf*9 expression to achieve 1:1 MR subunit stoichiometry. MR complexes were purified by NiNTA affinity and ion-exchange chromatography[17]. MR[lo] was expressed at considerably lower amounts compared to MR[wt] or MR[ho], indicating that MR oligomerization may stabilize the complex. The MRX complex was expressed using vectors for His-tagged Mre11, untagged Rad50, and FLAG-tagged Xrs2 and purified *via* NiNTA and FLAG affinity chromatography[45]. Recombinant Mre11 was purified by amylose and NiNTA affinity chromatography, while Xrs2 and Rad50[wt/ho/lo] were purified with FLAG affinity resin[21]. Exo1 was purified by FLAG affinity and ion-exchange chromatography[45]. Recombinant phosphorylated Sae2 (pSae2) was purified from *Sf*9 cells by amylose and NiNTA affinity chromatography in the presence of phosphatase inhibitors[19]. Yeast RPA was expressed from the p11d-scRPA vector (kind gift from M. Wold, University of Iowa) in *E. coli* (BL21 DE3 pLysS) and purified as described for human RPA[15].

**DNA substrates**. Oligonucleotides used in this study are listed in Supplementary Table 1. To prepare the DNA substrates (20, 31, 70, 100, and 197 bp) for DNA binding and endonuclease experiments, the respective oligonucleotides were labeled at the 3′-end with [$\alpha-32$P]dCTP (Perkin Elmer) and terminal transferase (New England BioLabs) following the manufacturer's recommendations. The enzyme was inactivated, and the reaction purified on Micro Bio-Spin P-30 Gel Columns (Bio-Rad). The labeled oligonucleotide was then annealed with twofold excess of the related unlabeled oligonucleotide in 1× PNK buffer (New England BioLabs). The 5′-labeled 70 bp substrate for exonuclease assays was prepared similarly, except that the PC210 oligonucleotide was labeled at the 5′-end with [$\gamma$−32P]ATP (PerkinElmer) and T4 polynucleotide kinase (New England BioLabs). Oligonucleotides for hot annealing (5′_OligoA and 3′_OligoA) were labeled at the 5′-end with [$\gamma$−32P]ATP (PerkinElmer) and T4 polynucleotide kinase (New England BioLabs) following the manufacturer's recommendations. For mass photometry, oligonucleotides of 20, 50, 73, 106, and 197 bp (Supplementary Table 1) and 2.7 kbp-long pUC19 were used. For TEM, circular 5.3 kbp-long

pFBDM (Addgene; Fig. 1b-iii and Supplementary Fig. 1f), linear 2.7 kbp-long pUC19 with free 4 nt-overhangs or a 2.8 kbp-long, pUC19-derived linear plasmid with streptavidin-blocked ends were used. Where indicated, circular, relaxed or linear 2.7 kbp-long pUC19 with free 4 nt-overhangs, and a 2.8 kbp-long, pUC19-derived blunt linear plasmid with streptavidin-blocked ends were used for EMSAs and nuclease assays. The circular, supercoiled form of pUC19 was relaxed with Topoisomerase I from *E. coli* (New England Biolabs) and purified using a MinElute PCR purification kit (Qiagen). Linearized pUC19 was prepared with restriction digestion using KpnI-HF (New England BioLabs) and 0.8% agarose-gel purification. The 2.8 kbp-long, pUC19-derived linear plasmid substrate with biotin at the ends was prepared using pAttP-S vector, annealed PC210 and PC211 oligonucleotides and ΦC31 integrase[68]. After incubation with monovalent streptavidin (kind gift from M. Howarth, University of Oxford), streptavidin-blocked 2.8 kbp-long pUC19 was purified with Chromaspin TE-200 columns (Takara). For Telomeric PCR, the 100 μM stocks of oligonucleotides listed in Supplementary Table 1 were diluted in sterile water to a working concentration of 10 μM. oBL359 was purchased from Microsynth; oBL358, oBL360, and oBL361 from Merck Millipore.

**Negative staining TEM**. Purified proteins were mixed at the indicated concentrations with or without plasmid substrates to 5, 10, or 15 μl reactions on ice in a reaction buffer containing 25 mM Tris acetate pH 7.5, 1 mM dithiothreitol (DTT) and the co-factors 5 mM magnesium acetate (Sigma-Aldrich, M5661), 1 mM manganese chloride (Sigma-Aldrich, M3634) and 1 mM ATPγS (Sigma-Aldrich, A1388), unless indicated otherwise. In Supplementary Fig. 1f-ii and for resection reactions in Fig. 2d, 1 mM ATP (Sigma-Aldrich, A2383) was used instead of ATPγS. KCl in phosphate buffer (storage buffer of MR; 20 mM phosphate solution pH 7.4 [$KH_2PO_4$, $K_2HPO_4$], 10% glycerol, 1 mM phenylmethylsulfonyl fluoride [PMSF], 1 mM DTT) was added to adjust salt concentrations as indicated. Assessment of oligomer size in Figs. 1a and 3d was performed at 120 mM KCl. For MR[wt/ho/lo]/MR[wt]-pSae2/Mre11[wt]/Rad50[wt/ho/lo] binding to plasmid DNA, 30 mM KCl was used for Fig. 1b and Supplementary Fig. 1e, f; 80 mM KCl for Fig. 1d-ii; 35 mM KCl for Fig. 4b; 50 mM KCl for Fig. 4c; 40 mM KCl for Supplementary Fig. 4c; otherwise specified in the Figure itself. The reactions were incubated at 30 °C for 30 min, unless indicated otherwise.

For Fig. 2d-i, I and 2d-ii, I, 5 nM 2.7 kbp-long, linear plasmid DNA (in molecules) was incubated for 2.5 h with 75 nM RPA at 30 °C and 40 mM KCl. For Fig. 2d-i, II, III, 50 nM MR[wt] and 200 nM pSae2 were added and incubated for 2.5 h at 30 °C with 40 mM KCl. For Fig. 2d-ii, II, 5 nM 2.7 kbp-long, linear plasmid DNA (in molecules) was incubated with 50 nM Exo1 and 200 nM RPA for 5 min at 30 °C and 50 mM KCl. For Fig. 2d-iii, I, 5 nM 2.8 kbp-long, linear plasmid DNA (in molecules) carrying two streptavidin blocks per end was incubated with 50 nM Exo1 and 75 nM RPA for 2.5 h at 30 °C and 40 mM KCl. For Fig. 2d-iii, II, 50 nM MR[wt] and 200 nM pSae2 were incubated with 5 nM of the substrate (in molecules) for 2.5 h at 30 °C and 40 mM KCl. For Fig. 2d-iii, III, IV, 50 nM Exo1 and RPA to a final concentration of 670 nM were added to the reaction with MR[wt]-pSae2 (Fig. 2d-iii, II) for 10 min at 30 °C and 50 mM KCl.

For TEM grid preparation, 5 μl of the reaction mixture were incubated for 1 min at room temperature on a carbon film 300 mesh copper grid (CF300-CU from Electron Microscopy Sciences) that was negatively glow discharged with the Emitech K100X glow discharge system for 45 s with 25 mA. After blotting away the excess sample with Whatman filter paper, the grid was washed twice in EM buffer (20 mM HEPES pH 7.55, 130 mM NaCl, 2% glycerol, 0.5 mM DTT), stained in two droplets of 2% uranyl acetate, and air dried. TEM micrographs were acquired with a FEI Morgagni 268 microscope at 100 kV using the Morgagni User Interface 3.0, iTEM 5.2 software and a CCD 1376 × 1032 pixel camera at different magnifications. All samples were imaged at least three times, and image analysis was performed using Fiji/ImageJ 1.52t, Excel 2016, and GraphPad Prism 9.

**EMSAs**. DNA binding experiments with oligonucleotide-based DNA were carried out in buffer containing 25 mM Tris acetate pH 7.5, 1 mM DTT, 0.25 mg/ml bovine serum albumin (BSA), 5 mM magnesium acetate, and 1 mM ATPγS. To keep the total DNA amount constant, 5 nM 20 bp, 3.22 nM 31 bp, 1.43 nM 70 bp, 1 nM 100 bp, and 0.51 nM 197 bp DNA were used, such that 100 nM (in base pairs) of 3′-labeled DNA substrate was incubated with the indicated amount of protein at 30 °C for 15 min in 15 μl reaction volume at 30 mM NaCl. Other indicated final salt concentrations were reached by NaCl addition. After incubation, the samples were supplemented with 5 μl loading dye (50% glycerol with bromophenol blue) and loaded on a 6% polyacrylamide gel (TAE). The gel was run for 90 min at 80 V (2.7 V/cm) on ice (75 min for the 20 bp substrate) in a Mini-PROTEAN® Tetra Cell system (BioRad). After separation, the gels were dried and exposed to a phosphor screen. The screen was then imaged using a Typhoon FLA 9000 imager with Typhoon FLA 9500 software (GE Healthcare), quantitated with Fiji/ImageJ 1.52t and Excel 2016, and statistically analyzed with GraphPad Prism 9.

For DNA binding experiments with plasmid-based DNA 2.5 nM (in molecules) circular 2.7 kbp-long Topoisomerase I-relaxed or KpnI-linearized pUC19 plasmid was used, as well as 2.5 nM (in molecules) streptavidin-blocked pUC19-derived 2.8 kbp substrate. Reactions of 20 μl were mixed in buffer containing 25 mM Tris acetate pH 7.5, 1 mM DTT, 80 mM KCl, 5 mM magnesium acetate, 1 mM manganese chloride, and 1 mM ATPγS. After incubation for 1 h at 30 °C, the reactions were mixed with TriTrack DNA loading dye 6× (Thermo Scientific[TM]) to

1× and loaded on a 0.5% agarose gel (TB). The gel was run for 1 h at 140 V (11.7 V/cm) on ice. After running, the gels were stained with GelRed Nucleic Acid Gel Stain (Biotium), imaged, and quantified as above.

**Nuclease assays.** Endonuclease assays (15 μl volume, unless indicated otherwise) were performed with 1 nM (in molecules) 70 bp-long DNA substrate labeled at the 3′-end in reaction buffer containing 25 mM Tris acetate pH 7.5, 1 mM DTT, 80 U/ml pyruvate kinase (Sigma-Aldrich), 1 mM phosphoenolpyruvate (PEP), 0.25 mg/ml BSA, 5 mM magnesium acetate, 1 mM manganese acetate, and 1 mM ATP. Monovalent streptavidin (30 nM) was added and incubated for 5 min at room temperature. The indicated proteins were added and incubated for 30 min at 30 °C. After incubation, the reaction was stopped with 0.5 μl of 14–22 mg/ml proteinase K (Roche), 0.5 μl of 10% (w/v) sodium dodecyl sulfate (SDS), and 0.5 μl of 0.5 M ethylenediaminetetraacetic acid (EDTA) and incubated at 50 °C for 30 min. For time-course experiments, proteins were added to a single master mix and incubated at 30 °C. Aliquots (15 μl) of the reaction were collected at the indicated time and stopped as described above. Each sample was mixed with equal volume of loading dye (95% formamide, 20 mM EDTA, 1 mg/ml bromophenol blue) and boiled for 4 min at 95 °C. The products were separated by denaturing polyacrylamide electrophoresis using a 15% polyacrylamide gel with 7 M urea. The gels were fixed for 30 min in fixing solution (40% methanol, 10% acetic acid, 5% glycerol) and then dried on 3MM paper (Whatman), exposed to phosphor screen, and imaged using a Typhoon FLA 9000 imager with Typhoon FLA 9500 software (GE Healthcare). The resulting images were analyzed with Fiji/ImageJ 1.52t, Excel 2016, and GraphPad Prism 9.

Exonuclease assays were performed with 1 nM (in molecules) 70 bp-long DNA substrate labeled at the 5′-end in reaction buffer containing 25 mM Tris acetate pH 7.5, 1 mM DTT, 0.25 mg/ml BSA, 5 mM manganese acetate, and 1 mM ATP or ATPγS where indicated. After protein addition, the reactions were incubated for 30 min at 30 °C, and subsequently stopped and processed as described above for the endonuclease assays.

Nuclease assays on plasmid-length substrate were performed with 1 nM (in molecules) 2.8 kbp-long, pUC19-derived substrate obtained with ΦC31 integrase (as described in the "DNA substrates" section) in reaction buffer as used for endonuclease assays containing 25 mM Tris acetate pH 7.5, 1 mM DTT, 80 U/ml pyruvate kinase, 1 mM PEP, 0.25 mg/ml BSA, 5 mM magnesium acetate, 1 mM manganese acetate, and 1 mM ATP. The substrate was incubated, where indicated, for 5 min at room temperature with 30 nM monovalent streptavidin. The indicated proteins were added and incubated at 30 °C. For Fig. 1d-i, the salt concentration was adjusted with KCl to 80 mM to match the TEM imaging conditions. At different time points (0, 15, and 30 min), 14 μl of the reaction were collected and transferred into tubes containing 1 μl of Exo1 or an equal amount of storage buffer and further incubated for 10 min at 30 °C. The samples without MR$^{wt/ho}$-pSae2 were collected from the master mix before the addition of MR$^{wt}$/MR$^{ho}$ and pSae2 and processed with or without Exo1 as described above. After the additional 10 min, the reactions were stopped by the addition of 5 μl of 2% STOP solution (30 mM EDTA, 2% SDS, 30% glycerol, and 1 mg/ml bromophenol blue) and 1 μl Proteinase K (14–22 mg/ml). The reactions were deproteinated at 37 °C for 1 h, then 15 μl of each sample were mixed with a threefold excess of the indicated 22 nt-long radio-labeled probe compared to the plasmid-length substrate (in molecules). For Figs. 1d-i, 2a,b, 4h and Supplementary Figs. 4a and 6b, c, the radio-labeled probe anneals 160 nt away from the 5′-DNA end where the 5′-terminated strand was resected. For Supplementary Figs. 4b and 6d, the radio-labeled probe anneals 160 nt away from the 3′-DNA end where the complementary strand was resected. The mix was incubated for 3 min at 60 °C in a water bath and subsequently cooled down overnight to room temperature. The annealing reactions were loaded on a 1% agarose gel. The gels were then dried on DE81 chromatography paper (Whatman) and processed as described above.

**Mass photometry.** Mass photometry measurements were carried out on a OneMP device (Refeyn Ltd). Glass coverslips (No. 1.5H thickness, 24 × 50 mm, VWR) were cleaned by sonication, first in isopropanol and then in deionized water for 15 min. After cleaning, the coverslips were dried under a clean stream of nitrogen. For measurements with the 2.7 kbp-long pUC19 plasmid, coverslips were coated with a 1% poly-lysine solution. For each measurement, a clean coverslip was placed onto the objective. For sample delivery, a silicone gasket (CultureWell$^{TM}$ Reusable Gasket, Grace Bio-Labs) with four wells was fixed on the surface of the coverslip. Before measurements, samples were incubated in buffer containing 25 mM Tris acetate pH 7.5, 1 mM DTT, 0.1 mM ATPγS, 5 mM magnesium acetate, and 1 mM manganese chloride at 30 °C for 30 min.

Mass photometry experiments with oligonucleotides in Figs. 1f, 3e and Supplementary Fig. 3d were performed at 25 mM KCl with 5 nM MR$^{wt/ho}$ or MRX and 27.7 nM 20 bp, 9.8 nM 50 bp, 6.7 nM 73 bp, 4.6 nM 106 bp, and 2.5 nM 197 bp DNA to keep the total DNA amount constant (100 nM base pairs). Peaks were detected and fitted with Gaussian functions where possible. For Supplementary Figs. 3c and 5b, 5 nM Mre11/Rad50/Xrs2/MR$^{wt/ho/lo}$/MRX were measured with or without 2.5 nM 197 bp-long oligonucleotide DNA at 25 mM KCl. For mass photometry experiments in Supplementary Fig. 5d without DNA, 10 nM MR$^{wt/ho}$ was incubated at 250 mM KCl or 25 mM KCl and 30 °C for 30 min with co-factors as afore-mentioned. Binding was monitored for 40s.

For Supplementary Figs. 3e and 5g with 2.7 kbp plasmid DNA, 20 nM MR$^{wt/ho}$/MRX was first incubated with 1 nM substrate (in molecules) for 30 min at 250 mM KCl and 30 °C and subsequently measured (labeled as 250 mM KCl, 30 min). MR$^{wt}$ (~460 kDa) and plasmid DNA (~1300 kDa) appeared mostly unbound, while MRX and MR$^{ho}$ formed oligomers without DNA beyond the detection limit such that less free MRX or MR$^{ho}$ dimers could be detected at ~550 or ~460 kDa, respectively. Secondly, the salt in the sample was diluted to 25 mM KCl, while keeping the protein and DNA concentrations constant, and the sample was immediately analyzed (labeled 25 mM KCl). Much less unbound MRX, MR$^{wt}$ or MR$^{ho}$ and plasmid DNA could be detected. For MRX and MR$^{ho}$, however, the decrease in unbound plasmid DNA was smaller than for MR$^{wt}$, indicating that they form larger oligomers per DNA molecule. For MR$^{wt}$, a tail of counts at high molecular weight >2500 kDa, corresponding to higher-order MR-DNA assemblies, could be detected. MRX and MR$^{ho}$, on the other hand, formed much larger nucleoprotein assemblies beyond the detection limit (smaller tail of counts >2500 kDa). Third, the sample with 25 mM KCl was incubated for 30 min at 30 °C and then measured (labeled 25 mM KCl, 30 min). For both MR$^{wt}$ and MR$^{ho}$, even fewer higher-order assemblies on DNA could be detected >2500 kDa, as they surpassed the detection limit, while for MRX the tail of counts >2500 kDa remained similar. Binding was monitored for 100 s.

For data acquisition, the gasket well was filled with 9 μl measuring buffer for adjustment of surface focusing. Afterwards, 1 μl of sample was added to the buffer in the well resulting in the final sample concentration. Protein and complex binding to the coverslip surface was monitored using AcquireMP (Refeyn Ltd, version 2.3.0). Data analysis was performed by DiscoverMP (Refeyn Ltd, version 2.3.0), OriginPro 2017 and a custom-written Python program (Supplementary Software). For contrast to mass conversion, a known protein mass standard calibrant (NativeMark$^{TM}$ Unstained Protein Standard; Invitrogen) was measured on the same day. Due to calibration with respect to protein mass, the pUC19 peak in plasmid measurements appeared at ~1300 kDa. All samples were measured at least three times.

**Thioflavin T (ThT) fluorescence.** For ThT measurements, 20 μl reactions of 1.2 μM MR$^{wt}$ or MR$^{ho}$ were incubated for 30 min at 4 or 42 °C in phosphate buffer containing 20 mM phosphate solution pH 7.4 (KH$_2$PO$_4$, K$_2$HPO$_4$), 10% glycerol, 1 mM PMSF, 1 mM DTT, and 240 mM KCl. Thioflavin T (ThT; Sigma-Aldrich, T3516) was dissolved in water to a final concentration of 2.5 mM and filtered (0.2 μm; Millipore). ThT solution was added (1:10 dilution) to the samples, and fluorescence intensity measured in a 384-well plate (Corning Life Sciences) using a CLARIOstar plate reader (BMG Labtech, 5.40 R3 software). Excitation wavelength was set at 450 nm and emission recorded at 490 nm. Statistical analysis was performed with GraphPad Prism 9.

**Pelleting assay.** For pelleting assays, 60 μl reactions of 1.2 μM MR$^{wt}$ or MR$^{ho}$ were incubated for 30 min at 4 or 42 °C in phosphate buffer as for ThT assays. After incubation, the reactions were centrifuged at 21,000g for 10 min at 4 °C. The resulting pellets were washed twice with phosphate buffer, resuspended in urea sample buffer (116 mM Tris-HCl pH 6.8, 5% glycerol, 8 M urea, 5% SDS, 1% beta-mercaptoethanol, 0.1% bromophenol blue) and loaded likewise to the supernatant fractions onto a denaturing 4–12% BisTris SDS-PAGE (Thermo Fisher Scientific) that was then Coomassie-stained. MR$^{wt/ho}$ bands were normalized in intensity to their input using Fiji/ImageJ 1.52t and Excel 2016; statistical analysis was performed with GraphPad Prism 9.

**pSae2-Rad50 protein interaction assay.** For interaction assays, 50 μl of soluble extract from *Sf*9 cells, expressing the MBP-tagged C-terminal part of pSae2 (pSae2 △N169)[19] with phosphatase inhibitors, were incubated with 50 μl Amylose resin (New England Biolabs) in 450 μl lysis buffer (50 mM Tris-HCl pH 7.5, 2 mM beta-mercaptoethanol, 1 mM EDTA, 15.6 % glycerol, 305 mM NaCl, 30 μg/ml leupeptin, 1 mM PMSF, and protease inhibitor cocktail (1:400, Sigma-Aldrich, P8340)) for 1 h at 4 °C with agitation. The resin was then centrifuged for 2000g for 2 min at 4 °C and washed five times with wash buffer containing 50 mM Tris-HCl pH 7.5, 150 mM NaCl, 0.2% NP40, 2 mM EDTA, and 1 mM PMSF in a total volume of 1.3 ml. After the addition of 1.5 μg purified MR$^{wt/ho/lo}$ and 250 μl wash buffer, the mix was further incubated for 1 h at 4 °C with agitation. MR$^{wt/ho/lo}$ controls were also incubated with amylose resin without pSae2 △N169. The resin was then centrifuged and washed five times as described above. MBP-tagged pSae2 △N169 bound to pulled-down MR$^{wt/ho/lo}$ via Rad50 interactions was eluted with 100 μl wash buffer containing 20 mM maltose (Carl Roth). The eluate (20 μl) was then loaded with 1× NuPAGE$^{TM}$ LDS Sample buffer on a denaturing 4–12% Bis-Tris SDS-PAGE (both Thermo Fisher Scientific). Western blots were performed with mouse α-MBP antibody (1:1000, Abcam, ab49923), rabbit α-Rad50 antibody (1:1000, ThermoScientific, PA5-32176), and goat α-rabbit or α-mouse HRP-conjugated secondary antibody (1:2000, BioRad, 170-6515 or 170-6516, respectively). The blots were imaged using a Vilber Fusion FX6 system and FusionCapt Adv FX7 software.

**Yeast plasmids and strains.** Construction of yeast plasmids (Supplementary Table 2) and strains (Supplementary Table 3) was performed according to standard

molecular biology methods. Yeast strains used in this study are derivatives of BY4741. Cells were grown at 30 °C in yeast extract pentose dextrose, YPD (2% glucose, 2% peptone, 1% yeast extract) or synthetic SD media (2% glucose, 0.17% yeast nitrogen base, 0.5% $NH_4$ sulfate, and amino acids). For galactose-induced overexpression experiments, overnight grown cells in SD media (2% raffinose) were diluted in SD 2% galactose or SD 2% glucose for 5 h before spotting assays or harvesting protein samples.

**DNA damage sensitivity spotting and growth assays**. Serial dilutions of exponentially growing cells were spotted on YPD or SD-URA (with 2% glucose or 2% galactose) agar plates without or with DNA damaging agents: Camptothecin, CPT 25 µM (Sigma-Aldrich, C9911) and Hydroxyurea, HU 25 mM (Sigma-Aldrich, H8627). Agar plates were incubated at 30 °C and imaged after 2 days using a Vilber Fusion FX6 system and FusionCapt Adv FX7 software. For growth rates, $OD_{600}$ of exponentially growing cells was measured in the absence or presence of CPT (5 or 25 µM) in a 24-well plate using a plate reader (CLARIOStar BMG Labtech, 5.40 R3 software). Doubling times for exponential growth were calculated using the following Eq. (1):

$$\text{Doubling time (hours)} = \frac{\text{duration (hours)} \times \log(2)}{\log(\text{final OD}_{600}) - \log(\text{initial OD}_{600})} \qquad (1)$$

Results from independent replicates ($n = 3$) were averaged and displayed using GraphPad Prism 9.

**Live-cell fluorescence microscopy and image analysis**. Yeast cells were grown overnight in SD media (2% glucose) and then diluted into fresh growth medium to resume exponential phase growth. Before imaging, cells were left untreated or treated with 200 µM CPT for 2 h. Live-cell microscopy was performed using a spinning disk confocal fluorescence microscope (Visitron system) equipped with an inverted microscope (Nikon Eclipse TiE), an EM-CCD camera (Andor iXon Ultra, Andor), a motorized XY stage, and a piezo Z drive. The system is controlled by VisiVIEW software 5.0.0 (Visitron system). Cells in 96-well plates were imaged at 30 °C in a temperature-controlled incubator with a ×100 objective lens (NA = 1.4, Nikon CFI Plan Apo). For image analysis, the percentage of cells containing Rad50 foci in the nucleus, based on histone Hta2-mCherry signal, was quantified using the Cell Counter plugin in Fiji/ImageJ 1.52t software. Results of independent biological replicates ($n = 3$) were averaged and displayed with GraphPad Prism 9. Furthermore, automated single-cell image analysis was performed using Yeast-Quant 2021 software[69] on raw images running in Matlab R2017b. Cells were segmented based on brightfield channel images and nuclei were segmented based on Hta2-mCherry. Relocalization and foci formation of Rad50-yEVenus were quantified as normalized standard deviation of yEVenus average intensity signal. At least 50 cells per condition were analyzed and quantified.

**Rad50/Rad53 western blots and flow cytometry analysis**. Total protein lysates were prepared according to standard procedures. Briefly, cells of $OD_{600} = 1-2$ were resuspended in 150 µl lysis solution (1.85 M NaOH, 1.09 M beta-mercaptoethanol) and precipitated in 50% TCA on ice for 10 min. After centrifugation, the pellet was resuspended with 1 ml acetone and centrifuged again. Then the pellet was resuspended in 100 µl urea buffer (120 mM Tris-HCl pH 6.8, 5% glycerol, 8 M urea, 143 mM beta-mercaptoethanol, 8% SDS, bromophenol blue) and loaded on a 7.5% acrylamide gel (Rad50 blot) or on a 7.5% TGX SDS-PAGE gel (BioRad) (Rad53 blot). The following primary antibodies were used for immunoblotting: α-Rad50 (1:1000; ThermoScientific, PA5-32176), α-Tubulin (1:20,000; Sigma-Aldrich T5168), α-Pgk1 (1:10,000; Invitrogen, 459250), and α-Rad53 (1:1000; Abcam, ab166859). Goat α-rabbit or α-mouse HRP-conjugated secondary antibody (1:2000; BioRad, 170-6515 or 170-6516 in Rad50 blot/170-5047 in Rad53 blot, respectively) were used as secondary antibodies. Blots were developed using the SuperSignal West Pico PLUS Chemiluminescence substrate (Thermo Fisher Scientific, 34579), and images were acquired using a Vilber Fusion FX6 system (FusionCapt Adv FX7 software).

For flow cytometry analysis, cells with $OD_{600} = 0.5$ were fixed in 70% ethanol overnight. Pellets were treated with RNase A (Thermo Fisher Scientific, 10753721) at 37 °C for 2 h and Proteinase K (Biofroxx, 1151ML010) at 50 °C for 2 h in 50 mM Tris-HCl pH 7.5 buffer. The cell suspension was sonified using a Branson sonifier 450 for 5 s with output control 1 and duty cycle constant. Then, cells were stained with 2.4 µM SYTOX Green (ThermoFisher Scientific, 1076273). Measurement was performed on the BD LSRFortessa flow cytometer (BD Biosciences) using the BD FACSDiva software (v9.0.1). In total, 20,000 events were recorded. Analysis was performed with FlowJo (v10.8.0) using the gating strategy as indicated in Supplementary Fig. 7b and the Reporting summary.

**Isolation of genomic DNA and Telomeric PCR**. Untagged strains were grown in YPD medium until the cultures reached the exponential growth phase and $OD_{600} = 0.8-1$. Then, 10 ml of the cell suspension were harvested for genomic DNA (gDNA) isolation using the Gentra Yeast/Bact. PureGene Kit (Qiagen) according to the yeast protocol section with the following minor modifications: 10 ml cells were used as input, 10 µl Zymolyase 20T (5 units/µl, amsbio 120491-1) were added and incubated for 1 h at 37 °C. The gDNA concentration was measured using the Nanodrop2000c Spectrophotometer (ThermoFisher). The isolated gDNA was stored at 4 °C.

Telomeric PCR was performed as described in Förstemann et al.[70] with minor modifications. All steps were carried out on ice and in the C1000 Touch Thermal Cycler (Bio-Rad). The C-tailing reaction (7.1 µl sterile water, 0.9 µl 10× NEBuffer 4 (New England Biolabs, B7004S), 1 µl of 100 ng/µl gDNA) was incubated for 10 min at 96 °C to denature the DNA, then the mix was cooled down to 4 °C. Next, 1 µl of 10× tailing mix (2 µl 2 units/µl Terminal transferase (New England Biolabs, M0315S), 1 µl 10× NEBuffer 4, 1 µl 10 mM dCTP (Thermo Fisher, R0151), 6 µl sterile water) was added and incubated for 30 min at 37 °C, 10 min at 65 °C, 5 min at 96 °C, and kept at 65 °C. Then, 30 µl of 65 °C preheated PCR-mix (21 µl sterile water, 4 µl 10× Telo-PCR buffer (670 mM Tris-HCl (pH 8.8), 160 mM $(NH_4)_2SO_4$, 50% glycerol, 0.1% Tween-20), 4 µl 2 mM dNTP mix (ThermoFisher, R0191), 0.3 µl 10 µM forward oligonucleotide (1L oBL358, 6Y' oBL361, or 6R oBL360), 0.3 µl 10 µM reverse oligonucleotide (G18 oBL359), and 0.4 µl Phusion High Fidelity DNA polymerase (2 U/µl) (New England Biolabs, M0530S)) were added to the C-tailing reaction. The distance of the subtelomeric oligonucleotides to the first TG repeat is 119 bp for 6Y' oBL361, 39 bp for 1L oBL358, and 88 bp for 6R oBL360. The Telo-PCR program was run with the following steps: initial denaturation 3 min 95 °C, 45 cycles: 30 s 95 °C, 15 s 63 °C, 20 s 72 °C, and a final extension of 5 min at 72 °C. Finally, 8 µl of 6× Ficoll orange loading dye (15% Ficoll Type 400, 10 mM EDTA pH 8.0, Orange G) were mixed with each Telo-PCR sample and the entire PCR product was separated on a 1.8% 1× TBE-agarose gel using 0.5× SYBR Safe DNA gel stain (ThermoFisher, S33102) and the 100 bp DNA ladder (N3231S, New England Biolabs). Gel images were taken with the ChemiDoc Touch Imaging System (BioRad) and the analysis of the mean telomere length and distribution was executed using the ImageLab software version 5.2.1 (BioRad). Graphs were generated using the Graph Pad Prism 9 software.

**Statistical analysis**. All data shown in this study are representative of at least three independent experiments if not stated otherwise in the figure legends. See respective figure legends and "Methods" section for details to plotted data points, error bars, and specific data analyses. Statistical unpaired, two-tailed significance t-tests were performed using GraphPad Prism 9. For Fig. 1e, the t-value was 12.63 with 5 degrees of freedom. For Fig. 3c, the t-value was 22.51 with 6 degrees of freedom. For Fig. 3f, the t-values were 3.37 (Pellet) and 3.05 (Sup.), each with 4 degrees of freedom. For Fig. 4e, the t-values were 2.95 (12.5 nM), 4.18 (25 nM), and 6.04 (50 nM), each with 4 degrees of freedom. For Fig. 5g, the t-value was 3.03 with 4 degrees of freedom. For the non-significant t-test in Supplementary Fig. 1c-iv, the t-value was 1.47 with 198 degrees of freedom. For Supplementary Fig. 4a, the t-values were 4.28 (**), 6.6 (***), and 22.86 (****) with 6 degrees of freedom each. For Supplementary Fig. 5e, the t-values were 3.56 (*), 6.25 (***), and 22.51 (****) with 6 degrees of freedom each.

**Reporting summary**. Further information on research design is available in the Nature Research Reporting Summary linked to this article.

## Data availability
The data supporting this study are available from the corresponding authors upon reasonable request. The Rad50 crystal structure (identifier 5DAC https://doi.org/10.2210/pdb5DAC/pdb) was obtained from the RCSB Protein Data Bank (PBD, https://www.rcsb.org/). Rad50 patient mutations were found in the ClinVar (https://www.ncbi.nlm.nih.gov/clinvar/) and MedGen (https://www.ncbi.nlm.nih.gov/medgen/) Databases. Source data are provided with this paper.

## Code availability
An exemplary custom-written Python code for the analysis of mass photometer data is provided with this study as a Supplementary Software file.

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

## Acknowledgements

We thank Marc Wold (University of Iowa, US) for the p11d-scRPA vector, Mark Howarth (University of Oxford, UK) for monovalent streptavidin, and Susan Gasser and Kenji Shimada (FMI, Basel) for Rad53 antibodies and protocols. We are grateful to Maryna Levikova-Denzler, Cosimo Pinto, Radoslav Enchev, Weaam Mohamed and Anne Schreiber for technical advice and assistance; Peter Tittmann, Stephan Handschin, Miroslav Peterek, ScopeM, and the ETH Cryo-EM Knowledge Hub for assistance and access to electron microscopy; the Institute of Molecular Biology (IMB gGmBH) Flow Cytometry Core Facility for support with flow cytometry and Thomas M. Wismer for help with ClinVar/MedGen database analysis. We thank Josef Jiricny and members of the Cejka and Peter laboratories for fruitful discussions, and Jana R. Kissling, Silvia Napolitano, and Alicia Smith for critical reading of the manuscript. Work in the Cejka laboratory profits from funding by the Swiss National Science Foundation (SNSF, 31003A_175444, 310030_205199), the Swiss Cancer League, the Helmut Horten Foundation and the European Research Council (ERC, 681-630), while the Peter laboratory was supported by the SNSF, the Swiss Cancer league and ETH Zurich. The work of the Seidel laboratory was supported by the European Research Council (ERC). Work in the Luke lab was supported by the Deutsche Forschungsgemeinschaft (DFG, German Research Foundation) Project-ID 3935478939-SFB1631 and the Heisenberg program of the DFG, LU 1709-2-1.

## Author contributions

V.M.K. purified proteins, designed the Rad50 mutants with structural support from R.G., developed the in vitro pathway reconstitution technique in negative staining TEM, and performed experimental design, TEM analysis, biochemical assays, statistical analysis, conceptualization and visualization. G.R. purified proteins, and conducted, analyzed, and conceptualized radioactive DNA binding and activity assays. E.B. designed the yeast experiments, carried out the quantitative live-cell imaging, and performed image analysis with the help of S.S.L.; E.B. and J.T. engineered the yeast strains, and conducted immunoblotting, growth measurements, spotting assays, and data analysis. K.K. and R.S. are responsible for the mass photometry analysis; G.C. carried out the ThT fluorescence measurements and pelleting assays; N.S. and B.L. conducted the telomeric PCR, Rad53 immunoblotting, and flow cytometry analysis. M.P. and P.C. designed, supervised, and conceptualized the study; V.M.K., P.C., and M.P. wrote the paper with critical input from all authors. K.K. and J.T. contributed equally.

## Competing interests

The authors declare no competing interests.
