## [Peer Review File · Nature Communications]

REVIEWER COMMENTS

Reviewer #1 (Remarks to the Author):

The Mre11-Rad50-Nbs1/Xrs2 (MRN/X) complex is a factor involved in the repair of DNA double-strand breaks. It has been observed for some time, mostly in atomic force microscopy studies, that the MRN/X complex forms foci at damaged DNA in living cells and molecular aggregates near DNA ends in vitro. The biological mechanism and function of the oligomerization is not understood and certainly a highly relevant topic in the biology of DNA double-strand break repair.

Using a variety of different techniques, Kissling et al. investigate the oligomerization of the budding yeast ScMre11-Rad50 complex. They investigate properties of DNA binding and DNA processing, latter using reactions that also contain additional factors Sae2, Exo1 and RPA. Using negative stain electron microscopy, the authors visualize formation and requirements of different higher order assemblies of the MR complex using negative stain EM, mass photometry and EMSA. A very nice experiment is the stepwise processing of DNA ends and subsequent analysis of the protein-DNA complexes by negative stain images. Based on a published crystal structure of the Rad50 ATP binding domain in complex with DNA, the authors identify from crystal lattice contacts a Rad50-Rad50 interface that may be involved in oligomer formation. They characterize two sets of mutants in this region that enhance or disrupt MR complex multimerization and its effect on the function of the MR complex. In negative stain EM, the ho mutant formed oligomers, whereas the lo mutant did not form multimers. Intriguingly, the ho mutant had a slightly enhanced endonuclease activity and the lo mutant was deficient. The in vivo relevance of the mutants was analyzed by yeast survival assays.

Overall, the biochemical data are mostly clear and, as expected for these laboratories, of high quality. The paper is well written and clear. I applaud the authors for the structural investigation of the resection reaction, this is a really nice advance. However, I do have some concerns with respect to the interpretation the observed interface and the main conclusions regarding the role the oligomerization reaction as critical for endonuclease in vivo. In my opinion, the authors need to provide more evidence that the interface functions indeed as they propose and rule out an alternative explanation for the nuclease effects.

My specific points are as follows:

- 1) The point I see most critical is the present interpretation of the lo/hi mutants. In bacterial MR complex (SbcCD), the Rad50/SbcC interface site under investigation here is bound at or bound nearby by Mre11/SbcD during DNA cleavage. While we do not know whether eukaryotic MRN/X has a similar conformation during nucleolytic reaction, it is at least plausible and probably also likely. In any case, in such an alternative scenario the mutants stabilize or destabilize the nucleolytic active conformation of MR. This would explain the striking effects of the mutants on the nuclease in vitro and of course could also explain the observed effects in vivo. In addition, oligomerization could involve the Mre11 subunit for that reason. rather than being based on Rad50-Rad50 interactions. As most of the conceptual advance hinges on the mechanism of oligomerization and its importance in resection, perhaps the authors could think of a suitable experiment to distinguish between both possibilities? I wonder if a demonstration that Rad50 w/o Mre11 oligomerizes on DNA and this behavior is affected by the mutants might work?
- 2) Why did the authors choose to work with MR and not MRX, as this is the physiological form of the complex? While Xrs2 is dispensable for many functions of the complex in vivo, it would still be important whether the oligomerization properties of MRX are similar to MR?
- 3) I am not sure whether "cooperativity" in its thermodynamic meaning can be deduce from

the EMSA assays since EMSAs are not an equilibrium method. In Fig. 1e the authors present KD values that were obtained from EMSAs shown in Supp Fig. 2a. Titration of MR shifts the DNA into the pockets of the gel, even when small DNAs that could only bind one MR complex were used. This indicates aggregation/clustering of the complexes, a process that is typically irreversible and thus renders the calculation of an equilibrium constant inaccurate. I also do not entirely understand how the authors deduce cooperative binding from the mass photometry experiments. Please explain better and perhaps be more precise with respect to thermodynamics?

Reviewer #2 (Remarks to the Author):

The MRX complex functions together with Sae2 in end resection and DSB signaling. Each component in the complex makes intra- and inter-molecular interactions with various cellular proteins, and these interactions are critical in regulating endo- and exonuclease activity and damage signaling. In this study, authors showed the DNA end resection processing in step-by-step using negative staining TEM and demonstrated that *S. cerevisiae* MR complex forms ordered oligomers through a conserved b-sheet, which promotes the endonuclease activity, foci formation and damage signaling. They designed two mutants at the b-sheet of Rad50, each of which forms a higher or lower form of oligomer with altered properties. Although oligomerization of the MR complex has been known for quite some time, it is not well-established how the complex oligomerizes at the atomic level and how the MRN/X oligomerization is linked to its cellular activity. In the present study, authors have attempted to address these important issues using various approaches. However, there are a few uncertainties in this work; in particular, the issues of the characterization of oligomeric states, two MR mutants, and clarity of negative stained (NS) images should be resolved to provide clear understanding to the readers.

1. Due to the resolution limit, many NS images are ambiguous and this reviewer cannot agree with the authors' interpretation on several images. These include Figs 1b (ii, iii), 2d (i: II, III and iii: II) (see below).

(i) Figure 1b ii and iii: (line 115- 119) I see many isolated MR dimers in this figure, and in fact, the isolated MR dimers are as many as the two closely positioned MR dimers. It is unclear if all MR dimer pairs are formed through the dimer-dimer interaction or each dimer is simply closely located without interaction. Can we conclude that these molecules interact each other in an ordered manner? It would be difficult to obtain high resolution images or a structure of the higher-order MR oligomer. However, to provide a more convincing model for an ordered dimer pair, authors should provide a high resolution image or a structure for two MR dimers.

(ii) Although I can see an oligomer that may be considered as "pearls-on-a-string" shape (line 143) on top right of Fig 1bii (and bottom of Fig S 1e), other oligomers are better to be defined as clusters of the MR complex. The "pearls-on-a-string" requires a close-up view with higher resolution image.

(iii) Fig 1b: i bottom (line 124-126) The figure clearly shows DNA loops but it does not justify the distant pairs of DNA into close proximity (authors should mark the tethered MR complex in the figure 1b).

(iv) Fig 1d ii shows oligomeric clusters of MR at higher MR ratio (200 nM) (line 141-142). What is the population of this oligomer in the image.

(v) Fig 2d I (II & III) and iii (II and III) are hard to follow. At this resolution, it is difficult to visualize the image as shown in cartoons (line 197-198). In fact, the MR orientation and interactions in the cartoon might mislead the readers. In many NS images (if not all), it would be helpful to readers if authors provide the cartoons in the same orientation of the molecules.

2. It is unclear why the authors omitted Xrs2 in this study. While Xrs2 does not participate in end resection, the protein directly interacts with DNA and many other proteins, and is involved in the signaling. The oligomerization of the complex (the central feature in this study) might be affected by the presence of Xrs2. The authors should show the effect of Xrs2 in oligomerization of the complex.

3. Fig 3. Authors used crystal packing information of the Ct Rad50-DNA structure (5DAC) to understand the interactions between the MR dimers. Crystal packing information cannot be considered as an accurate information for protein – protein interactions because it is derived from the controlled precipitation of proteins under very specific conditions. Importantly, there are many symmetry related molecules that interact with CtRad50. It is unclear how authors can conclude the interaction between the MR dimers in NS images is achieved in side-by-side packing as shown in Fig 3a. In addition, the characterization of the two mutants have following problems.

(i) In 5DAC (pdb), Arg126 from symmetry-related Rad50 is within ionic interaction distance from Glu125, which could promote the interactions of the two Rad50 molecules (Fig 3a and line 250-252). Authors completely ignored this important interaction, and describe the Glu125 (or Asp124 Sc Rad50) – Glu125 repulsive effect. Two Glu125 are near 5 Å apart. This should be reexamined.

(ii) Asn123 is near the Glu1124 and Glu1117 of different symmetry-related Rad50 (coiled-coil region). There are several additional interactions, and thus, one cannot conclude that the interactions between the b sheets are critical in oligomerization. It is clearly possible that the b sheet of Rad50 participates in the interactions, but the partner can be various parts of the MR head and even coiled-coil or other molecule such as Xrs2 if it is present in the complex.

(iii) The authors made Ala mutant for two polar groups, which leads to higher-order (ho) oligomerization. Do authors believe that the oligomeric nature of the wild-type and ho mutant are same? The salt concentration dependency and size of oligomers suggest that they form different types of interactions. If they differ in oligomeric state, what is the evidence that the MR dimers oligomerize in a “pearls-in-a string” shape first, which subsequently form a cluster in ho mutant? Also, how this worm-like ho MR oligomer exhibited better endonuclease activity?

(iv) The Rad50 mutant in lo MR: how do we know if the mutation effect is due to other factors (such as disruption of the local structure and/or failure to interact with others) rather than lack of oligomerization. The Mre11 binding in the gel can be observed in the partly disrupted MR complex. In lo mutant, the quadruple mutation is likely to disrupt the Rad50 structure, part of the Mre11 interaction and DNA binding as they are not distant from the DNA binding site. Perturbation of the local structure could easily affect the DNA binding and nuclease activity – ATP-dependent endonuclease activity can be affected. (Line 277, FigS5c i). Thus, this reviewer believes that the ATP_rS binding should be more clearly justified for quadruple Rad50 mutant and the decreased DNA binding, endonuclease activities and other effects of lo MR oligomer is due to the structural perturbation rather than failure of oligomerization.

(v) They used the structure of Ct Rad50-DNA complex (5DAC). However, this is a coiled-coil truncated structure. Several studies proposed that the DNA interacts with coiled-coil region of Rad50. Xrs2 and Mre11 also interacts with DNA, and thus the DNA-binding by the complex is not as simple as the authors have proposed.

4. Oligomeric states of the MR complex are not well characterized. Authors claim that the wild-type MR dimers oligomerize into “non-aberrant aggregates”. If the MR dimers assemble into ordered forms of oligomers, what are their overall shape?

(i) Which forms of the oligomer enhance the endonuclease activity – as there are various oligomeric forms.

(ii) How various oligomers (including the worm-like lo MR oligomer) interact with pSae2 ?

Response to referees for Kissling *et al.*

We would like to thank the reviewers for their interest and effort taken to comment on our manuscript.

General revisions:

We have introduced the following major modifications by including additional data:

1. Two reviewers inquired about the oligomerization properties of MRX and the reason for focusing on MR rather than the physiological MRX complex. We opted for MR to reduce complexity, and because we could not purify high amounts of MRX. Moreover, consistent with previous reports (doi: 10.1016/s1097-2765(01)00388-4; doi: 10.1016/j.csbj.2020.05.013), we found that MR dimers co-purify with 1 or 2 Xrs2 subunits, resulting in heterogeneity. We note that in yeast, the Xrs2 subunit is dispensable for DNA end resection (doi: 10.1016/j.molcel.2016.09.011), justifying the omission of Xrs2. Nevertheless, we now added an entirely new Figure (Supplementary Fig. 3), demonstrating that MR and MRX oligomerize comparably in EM (a), that MRX binding is also stabilized on longer DNA as MR in gel shifts and mass photometry (b-d) and that MRX oligomerizes on plasmid-length DNA (e). Finally, we would like to highlight that all our *in vivo* experiments (further extended in the revised version) were performed with yeast cells expressing wild-type Mre11 and Xrs2, thus validating the Rad50^{wt/ho/lo} variants in a physiological complex.
2. As suggested, we performed additional experiments to provide higher resolution electron micrographs and adapted the cartoons to match the EM images and enhance clarity (mainly pathway reconstitution in revised Fig. 2d).
3. A key point raised by both reviewers was whether the beta-sheet mutations affect the oligomerization of Rad50 directly, or whether the effect is rather indirect *via* e.g. Mre11. To directly address this point, we individually purified the Mre11 and Rad50^{wt/ho/lo} subunits and observed that the ho/lo mutations affected oligomerization of Rad50 (new Fig. 4c): while Rad50^o did not form oligomers, Rad50^{wt/ho} oligomerized similarly to MR^{wt/ho} on linear plasmid-length DNA. Moreover, we show that Mre11 alone does not oligomerize on DNA, demonstrating that the observed MR oligomerization property is indeed largely dependent on Rad50 and independent of Mre11.
4. We overexpressed the Rad50 variants on top of endogenous Rad50^{wt} with a galactose-inducible promoter in yeast cells. Interestingly, we observed that Rad50^{lo}-overexpressing cells showed impaired growth and higher sensitivity to DNA damage (revised Fig. 5, new panels c and d). This result suggests that Rad50^{lo} is not simply a misfolded or “dead” protein, but capable of acting in a dominant-negative manner to inhibit the endogenous MRX^{wt} complex.
5. We have further expanded our *in vivo* validation of the Rad50 mutants to demonstrate the importance of the Rad50 beta-sheet-mediated MR oligomerization, including checkpoint signaling (new Fig. 6a) and telomere maintenance (new Fig. 6b,c).

Detailed point-by-point answers to the specific reviewer comments.

Reviewer #1 (Remarks to the Author):

*Overall, the biochemical data are mostly clear and, as expected for these laboratories, of high quality. The paper is well written and clear. I applaud the authors for the structural investigation of the resection reaction, this is a really nice advance. However, I do have some concerns with respect to the interpretation the observed interface and the main conclusions regarding the role the oligomerization reaction as critical for endonuclease *in vivo*. In my opinion, the authors need to provide more evidence*

that the interface functions indeed as they propose and rule out an alternative explanation for the nuclease effects.

My specific points are as follows:

1) The point I see most critical is the present interpretation of the lo/hi mutants. In bacterial MR complex (SbcCD), the Rad50/SbcC interface site under investigation here is bound at or bound nearby by Mre11/SbcD during DNA cleavage. While we do not know whether eukaryotic MRN/X has a similar conformation during nucleolytic reaction, it is at least plausible and probably also likely. In any case, in such an alternative scenario the mutants stabilize or destabilize the nucleolytic active conformation of MR. This would explain the striking effects of the mutants on the nuclease *in vitro* and of course could also explain the observed effects *in vivo*. In addition, oligomerization could involve the Mre11 subunit for that reason rather than being based on Rad50-Rad50 interactions. As most of the conceptual advance hinges on the mechanism of oligomerization and its importance in resection, perhaps the authors could think of a suitable experiment to distinguish between both possibilities? I wonder if a demonstration that Rad50 w/o Mre11 oligomerizes on DNA and this behavior is affected by the mutants might work?

Answer: We agree with the reviewer. As suggested, we examined the oligomerization behavior of purified Rad50 wild-type and the ho/lo mutants alone. As expected, we observed that the mutations affect Rad50 oligomerization even in the absence Mre11. Interestingly, in contrast to Rad50^{wt} and Rad50^{ho}, Mre11 and the Rad50^{lo} mutant were unable to form oligomers on linear DNA (new Fig. 4c). Moreover, we included Thioflavin T (ThT) measurements to show the formation of larger beta-sheet structures for MR^{ho} compared to MR^{wt} at lower (4°C) and especially at higher temperature (42°C) that promotes oligomerization (Supplementary Fig. 5e). MR^{lo} could unfortunately not be tested in these ThT assays, which require large volumes and higher concentrations impossible to achieve with MR^{lo}. Of note, Mre11 is wild-type in these MR^{wt} and MR^{ho} preparations, corroborating that the extent of beta-sheet structure formation is largely governed by Rad50^{wt/ho}.

While these data demonstrate that Mre11 does not drive the observed oligomerization behavior, we cannot unequivocally prove that the mutated beta-sheet motif is directly mediating Rad50 oligomerization. Moreover, we cannot rigorously exclude that the Rad50 mutations alter the Mre11 conformation, and thus disrupted oligomerization may also affect the nuclease activity. We have thus adapted the revised discussion to introduce this limitation.

2) Why did the authors choose to work with MR and not MRX, as this is the physiological form of the complex? While Xrs2 is dispensable for many functions of the complex *in vivo*, it would still be important whether the oligomerization properties of MRX are similar to MR?

Answer: As suggested (please also note our response to general revisions point 1 above), we now include experiments with purified MRX (new Supplementary Fig. 3). Importantly, MR and MRX complexes oligomerize comparably in solution and on plasmid-length DNA. Moreover, like MR, MRX binding is stabilized on longer DNA as assayed by gel shifts and mass photometry. Together with the *in vivo* experiments performed with yeast cells expressing Xrs2, we conclude that MR and MRX oligomerization is comparable.

3) I am not sure whether "cooperativity" in its thermodynamic meaning can be deduce from the EMSA assays since EMSAs are not an equilibrium method. In Fig. 1e the authors present KD values that were obtained from EMSAs shown in Supp Fig. 2a. Titration of MR shifts the DNA into the pockets of the gel, even when small DNAs that could only bind one MR complex were used. This indicates aggregation/clustering of the complexes, a process that is typically irreversible and thus renders the calculation of an equilibrium constant inaccurate. I also do not entirely understand how the authors

deduce cooperative binding from the mass photometry experiments. Please explain better and perhaps be more precise with respect to thermodynamics?

Answer: We thank the reviewer for this comment and agree that our methods are not suitable to measure positive cooperativity in its thermodynamic sense. Indeed, in the case of MR binding to DNA, the binding affinity of adjacent MR molecules may not be enhanced. We propose that MR dimers binding together as an oligomer are more stable on DNA, without necessarily invoking higher affinity of the individual dimers. We have now rephrased the respective sections in the manuscript and avoid the term "cooperative".

We believe that single MR dimers or two adjacent MR dimers on DNA are intermediate states and represent a nucleation point that leads to fast accumulation of MR molecules at the adjacent region of DNA. Indeed, we rarely observe only three adjacent MR dimers by EM analysis and similarly, EMSA measurements with plasmid-length DNA show that the bound species rapidly shift to the wells in a narrow MR concentration range, indicating a switch-like clustering of MR on the DNA (Fig. 1c). In mass photometry, which is an equilibrium method with oligomers staying in solution, the following trend was observed with increasing oligonucleotide length: up to 73 bp, more and more DNA molecules were bound by one MR dimer each, resulting in a growing peak at 450-550 kDa. Between 73 bp and 106 bp-long DNA, this 450-550 kDa peak decreased and instead a new peak around 1000 kDa, representing two MR dimers per DNA molecule, and much larger MR-DNA assemblies beyond the detection limit of the instrument were formed. This recapitulates the trend of better MR binding to longer DNA seen in EMSAs with oligonucleotides (Fig. 1e) and indicates the stabilizing effect of oligomerization on MR binding to DNA. Moreover, when we added DNA to MR oligomers in EM experiments, we observed that they dissociated into smaller complexes including dimers (Fig. 1 and Supplementary Fig. 1), demonstrating that the large oligomeric MR assemblies are not irreversible aggregates. Importantly, MR oligomers are able to resect DNA ends (Fig. 1d), further supporting that they are enzymatically active, functional species.

Reviewer #2 (Remarks to the Author):

*The MRX complex functions together with Sae2 in end resection and DSB signaling. Each component in the complex makes intra- and inter-molecular interactions with various cellular proteins, and these interactions are critical in regulating endo- and exonuclease activity and damage signaling. In this study, authors showed the DNA end resection processing in step-by-step using negative staining TEM and demonstrated that *S. cerevisiae* MR complex forms ordered oligomers through a conserved β -sheet, which promotes the endonuclease activity, foci formation and damage signaling. They designed two mutants at the β -sheet of Rad50, each of which forms a higher or lower form of oligomer with altered properties. Although oligomerization of the MR complex has been known for quite some time, it is not well-established how the complex oligomerizes at the atomic level and how the MRN/X oligomerization is linked to its cellular activity. In the present study, authors have attempted to address these important issues using various approaches. However, there are a few uncertainties in this work; in particular, the issues of the characterization of oligomeric states, two MR mutants, and clarity of negative stained (NS) images should be resolved to provide clear understanding to the readers.*

1. Due to the resolution limit, many NS images are ambiguous and this reviewer cannot agree with the authors' interpretation on several images. These include Figs 1b (ii, iii), 2d (i: II, III and iii: II) (see below).

Answer: As suggested, we replaced a number of EM images with better resolution micrographs (see also general revisions point 2 above). Moreover, to better guide the reader, we now provide cartoons that more closely match the EM images.

(i) Figure 1b ii and iii: (line 115- 119) I see many isolated MR dimers in this figure, and in fact, the isolated MR dimers are as many as the two closely positioned MR dimers. It is unclear if all MR dimer pairs are formed through the dimer-dimer interaction or each dimer is simply closely located without interaction. Can we conclude that these molecules interact each other in an ordered manner? It would be difficult to obtain high resolution images or a structure of the higher-order MR oligomer. However, to provide a more convincing model for an ordered dimer pair, authors should provide a high resolution image or a structure for two MR dimers.

Answer: As spotted by the reviewer, single or two adjacent MR dimers can be seen dispersedly bound on DNA to similar extents (see Figure R1). We therefore adapted the revised text to point this out. However, the majority of MR molecules form “pearls-on-a-string”-like accumulations that are clearly non-randomly distributed on DNA. We have now added zoom-insets of these structures (revised Fig. 1b and corresponding cartoons), which show that the MR dimers are bound on DNA either adjacently, or with loops between them. Such an arrangement is anticipated from our previous biochemical study, where we observed stepwise cleavage of 5'-terminated DNA (see model in doi: 10.1073/pnas.1820157116). We note that the term "ordered manner" is somewhat ambiguous in this context, as it might implicate repeated units along the DNA (i.e. a polymer), which is clearly not the case with such highly heterogeneous species. Nevertheless, the simultaneous imaging and biochemical nuclease assays allow us to conclude that the observed MR oligomers are active species.

The reviewer is correct that we can not visualize the exact interaction interface of the MR units due to the low resolution of the TEM images. However, the non-random distribution of the MR dimers on the DNA substrates strongly suggests that the MR molecules indeed interact with each other on DNA to form oligomers. In the revised manuscript, we now provide additional evidence that the oligomer formation is indeed driven by Rad50 interactions and independent of Mre11 (see above general revisions point 3, new Fig. 4c).

Figure R1: Counting of MR dimers bound to DNA (in Figure 1b,ii). 25 single MR dimers were identified (left image), 12 MR dimers observed as adjacent pairs on DNA (middle) and at least 71 molecules were counted in “pearls-on-a-string”-like accumulations (right). The latter is likely an underestimation as the single MR dimers in “pearls-on-a-string”-like accumulations can overlap and hide each other. We have thus not performed such quantifications of individual images.

We indeed attempted to obtain high-resolution data of adjacent MR dimers on DNA with cryo-EM, but the sample was unfortunately too heterogeneous. Moreover, the molecules displayed significant flexibility, which additionally deteriorated the quality of the 2D classification (see Figure R2). We hope that the reviewer agrees that a high-resolution cryo-EM structural analysis exceeds the scope of this paper, as multiple teams attempted and failed in this effort.

Figure R2: Cryo-EM 2D classes of MR^{wt} incubated with 70 bp-long DNA on ice for 30 min with 1 mM ATPγS, 5 mM Mg²⁺, 1 mM Mn²⁺ and 0.05 % NP40S detergent (to reduce extensive oligomerization and tethering) in Tris-HCl buffer pH 7.5. While the 2D classes could hint at MR dimers bound to DNA, the heterogeneity and structural flexibility prevent high-resolution analysis.

(ii) Although I can see an oligomer that may be considered as “pearls-on-a-string” shape (line 143) on top right of Fig 1bii (and bottom of Fig S 1e), other oligomers are better to be defined as clusters of the MR complex. The “pearls-on-a-string” requires a close-up view with higher resolution image.

Answer: To address this point, we have included zoom-insets with matching cartoons (revised Fig. 1b) to better illustrate the “pearls-on-a-string” structures. At higher MR to DNA ratios, numerous MR dimers are bound to the DNA molecules so that the individual MR dimers can no longer be readily distinguished. As noted above, we observed previously that MR resects 5'-terminated DNA in a stepwise manner (doi: 10.1073/pnas.1820157116). This concept is strongly supported by the data presented here (most notably Fig. 2c), which together clearly demonstrate sequential endonucleolytic cuts. We thus believe that MR oligomerization at DNA ends likely facilitates the stepwise nucleolytic DNA cleavage.

(iii) Fig 1b: i bottom (line 124-126) The figure clearly shows DNA loops but it does not justify the distant pairs of DNA into close proximity (authors should mark the tethered MR complex in the figure 1b).

Answer: We have removed the respective panel to focus Fig. 1b on MR oligomerization on DNA. The newly introduced zoom-insets to the “pearls-on-a-string” structures and matching cartoons show DNA loops formed in between the DNA-bound MR molecules. Thereby, head domain interactions and Rad50 coiled-coil tethering of MR oligomers can generate the observed DNA loops and substrate compaction and bring distant parts of DNA into closer proximity, as illustrated in the revised Fig. 1g.

(iv) Fig 1d ii shows oligomeric clusters of MR at higher MR ratio (200 nM) (line 141-142). What is the population of this oligomer in the image.

Answer: We found that the MR-DNA assemblies observed at higher MR:DNA ratios are heterogeneous in size (depending on the extent of substrate compaction and the number of MR/DNA molecules clustered), but clearly different from “pearls-on-a-string”-like structures. The large MR oligomers on DNA are the predominant species at these higher MR:DNA ratios (see Figure R3), and the micrograph presented in the manuscript is thus a representative image. Please note that the corresponding biochemical data indicate that these clusters are nucleolytically active MR species.

Figure R3: Collection of representative micro-graphs of 200 nM MR^{wt} incubated with 2.5 nM linear DNA as in Fig. 1d,ii. **Left:** Field of view image of the cropped panel shown in Fig. 1d,ii. **Right:** Another representative image showing more MR clusters formed under these experimental conditions. Scale bar: 200 nm.

(v) Fig 2d I (II & III) and iii (II and III) are hard to follow. At this resolution, it is difficult to visualize the image as shown in cartoons (line 197-198). In fact, the MR orientation and interactions in the cartoon might mislead the readers. In many NS images (if not all), it would be helpful to readers if authors provide the cartoons in the same orientation of the molecules.

Answer: We thank the reviewer for this comment. As suggested, we replaced critical EM images with higher resolution micrographs and adapted the cartoons to better illustrate the corresponding EM images. Due to the flexibility of the molecules and the Rad50 coiled-coils the orientation of the MR complexes could in some cases be different than drawn in the cartoons (not always easily detectable).

2. It is unclear why the authors omitted Xrs2 in this study. While Xrs2 does not participate in end resection, the protein directly interacts with DNA and many other proteins, and involved in the signaling. The oligomerization of the complex (the central feature in this study) might be affected by the presence of Xrs2. The authors should show the effect of Xrs2 in oligomerization of the complex.

Answer: As suggested by both reviewers, we have now included new data comparing the oligomerization of MR and MRX complexes (new Supplementary Fig. 3). A detailed response to this important addition is provided in the description of our general revisions above (point 1). Briefly, the new experiments demonstrate that *in vitro*, purified MRX oligomerizes similarly to MR, consistent with our *in vivo* experiments analyzing the Rad50^{wt/ho/lo} variants in yeast cells expressing endogenous Xrs2 (and Mre11).

3. Fig 3. Authors used crystal packing information of the Ct Rad50-DNA structure (5DAC) to understand the interactions between the MR dimers. Crystal packing information cannot be considered as an accurate information for protein – protein interactions because it is derived from the controlled precipitation of proteins under very specific conditions. Importantly, there are many symmetry related molecules that interact with CtRad50. It is unclear how authors can conclude the interaction between the MR dimers in NS images is achieved in side-by-side packing as shown in Fig 3a.

Answer: We agree with the reviewer that crystal packing information is not ideal to retrieve information on protein-protein interactions. Nevertheless, in the absence of a high-resolution cryo-EM structure of two adjacent MR dimers on DNA, the crystal packing information allowed us to generate a plausible and testable hypothesis for how two MR dimers could potentially interact. To better address this point, we have modified the text to highlight how the data from Fig. 3a were used as a hypothetical model to guide the mutational validation.

In addition, the characterization of the two mutants have following problems.

(i) In 5DAC (pdb), Arg126 from symmetry-related Rad50 is within ionic interaction distance from Glu125, which could promote the interactions of the two Rad50 molecules (Fig 3a and line 250-252). Authors completely ignored this important interaction, and describe the Glu125 (or Asp124 Sc Rad50) – Glu125 repulsive effect. Two Glu125 are near 5 Å apart. This should be reexamined.

Answer: We thank the reviewer for pointing this out. We are aware that there is a salt-bridge between Arg126 of one Ct Rad50 molecule (Arg125 in Sc) and Glu125 of the other Ct Rad50 molecule (Asp124 in Sc) that is broken in the MR^{ho} mutant. However, the two Glu125 residues could come closer than 5 Å in solution and the alleviation of their repulsive force may drive the enhanced interactions between the MR^{ho} molecules (Fig. 3d for example), negating the effect of the lost salt bridge that would instead decrease the interactions between MR^{ho} molecules. Consistent with reduced electrostatic interactions, we show that the interactions between MR^{ho} molecules are less salt-sensitive. Based on these considerations, we propose that alleviation of the repulsive force is the main driver of the increased MR^{ho}-MR^{ho} interactions. However, we now mention the indicated salt bridge in the revised manuscript.

(ii) *Asn123 is near the Glu1124 and Glu1117 of different symmetry-related Rad50 (coiled-coil region). There are several additional interactions, and thus, one cannot conclude that the interactions between the β sheets are critical in oligomerization. It is clearly possible that the β sheet of Rad50 participates in the interactions, but the partner can be various parts of the MR head and even coiled-coil or other molecule such as Xrs2 if it is present in the complex.*

Answer: We agree that there may be other interactions beyond the beta-sheet that could contribute to oligomerization. However, our new data (see above general revisions point 3, new Fig. 4c) demonstrate that the oligomerization behavior is mediated by Rad50 and not Mre11. Moreover, the newly introduced Thioflavin T measurements in Supplementary Fig. 5e show larger structures formed by Mre11-Rad50^{ho} *via* beta-sheets compared to Mre11-Rad50^{wt}. The newly added Supplementary Fig. 3 shows that with the Xrs2 subunit present, MR^{wt} oligomerization is even more pronounced, although Xrs2 is not necessary to induce MR^{wt} oligomerization.

(iii) *The authors made Ala mutant for two polar groups, which leads to higher-order (ho) oligomerization. Do authors believe that the oligomeric nature of the wild-type and ho mutant are same? The salt concentration dependency and size of oligomers suggest that they form different types of interactions. If they differ in oligomeric state, what is the evidence that the MR dimers oligomerize in a “pearls-in-a string” shape first, which subsequently form a cluster in ho mutant? Also, how this worm-like ho MR oligomer exhibited better endonuclease activity?*

Answer: Indeed, mass photometry analysis suggests that MR^{ho} is less salt-sensitive than MR^{wt} (Supplementary Fig. 5d). Likewise, ThT measurements show that MR^{ho} forms larger beta-sheet structures compared to MR^{wt} (Supplementary Fig. 5e). As Mre11 was not modified, we conclude that the diminished electrostatic and increased beta-sheet interactions between MR^{ho} molecules originate from the specific ho mutations. Despite this difference, rare intermediate MR^{ho}-DNA structures are observed, in which single MR^{ho} dimers assembling next to each other can be distinguished (Figure R4). These rare intermediates then “grow” into the large and dense “worm”-like structures shown in Fig. 4b. Thus, we suggest that similarly to MR^{wt}, MR^{ho} initially forms “pearls-on-a-string”-like arrangements before larger assemblies are built *via* beta-sheet interactions and additional Rad50 coiled-coil tethering.

Figure R4: Collection of representative micrographs of MR^{ho} incubated with linear DNA (white arrows) showing intermediate structures in which single MR dimers can be distinguished. Scale bar: 100 nm.

We observed “worm”-like structures with 90 nM MR^{ho} on 2.5 nM linear plasmid DNA (Fig. 4b). Our biochemical experiments show that increasing MR^{ho} to 200 nM leads to an additional increase in resection activity, implying that the “worm”-like structures are nucleolytically active (Fig. 4h). It is possible that the conformational changes required for endonuclease activation are facilitated in the multimolecular assemblies. Figure R5 shows MR^{wt} and MR^{ho} nuclease activity up to 400 nM protein concentration (Fig. 1d and 4h terminate at 200 nM). It is apparent that the MR complexes are active even at very high concentrations when secondary DNA cleavage events were observed (resulting in smaller products).

Figure R5: Representative resection assays of $MR^{wt/ho}$ with a probe (green box in cartoon) that binds to the 3'-overhang produced in resection as in Figures 1d,i and 4h.

(iv) The *Rad50* mutant in *lo MR*: how do we know if the mutation effect is due to other factors (such as disruption of the local structure and/or failure to interact with others) rather than lack of oligomerization. The *Mre11* binding in the gel can be observed in the partly disrupted MR complex. In *lo* mutant, the quadruple mutation is likely to disrupts the *Rad50* structure, part of the *Mre11* interaction and DNA binding as they are not distant from the DNA binding site. Perturbation of the local structure could easily affect the DNA binding and nuclease activity – ATP-dependent endonuclease activity can be affected. (Line 277, FigS5c i). Thus, this reviewer believes that the ATP_{RS} binding should be more clearly justified for quadruple *Rad50* mutant and the decreased DNA binding, endonuclease activities and other effects of *lo MR* oligomer is due to the structural perturbation rather than failure of oligomerization.

Answer: The reviewer is correct that we cannot rigorously exclude that the $Rad50^{lo}$ structure is affected by the introduced mutations, and that the mutations might impair the function of the complex beyond disrupting oligomerization. We now explicitly mention this limitation in the revised manuscript. However, we point out that (i) the untagged $Rad50^{lo}$ subunit can still be pulled down *via* tagged $Mre11^{wt}$, thus the complex is sufficiently stable under these conditions (Supplementary Fig. 5a), (ii) MR^{lo} retains exonuclease activity (Supplementary Fig. 6e), (iii) MR^{lo} is able to functionally interact with ATP (Supplementary Fig. 6e), (iv) MR^{lo} exhibits only a minor defect in DNA binding (Fig. 4a) and (v) MR^{lo} efficiently binds pSae2 (Fig. 4i). Notably, the latter interaction differentiates the $Rad50^{lo}$ mutant from *Rad50S* mutants, which fail to interact with pSae2 (doi: 10.1038/s41467-018-06417-5; doi: 10.1016/0092-8674(90)90524-i). Finally, MR^{lo} in yeast cells can still bind Xrs2 as it localizes to the nucleus (Xrs2 is required for nuclear localization; doi: 10.1016/j.molcel.2016.09.011; Fig. 5e). That said, we noted slightly lower expression levels of $Rad50^{lo}$ compared to $Rad50^{wt}$ (Fig. 5f), as also mentioned in the text.

However, the strongest support that MR^{lo} is not simply a misfolded and inactive complex stems from the new data presented in the revised Fig. 5c,d. We now show that $Rad50^{lo}$ overexpression causes a dominant-negative effect in yeast cells even without exposure to HU or CPT, thus interfering with the function of endogenous $MR^{wt}X$ complexes (see comments above, in particular general revisions point 4). These results suggest that increased levels of $MR^{lo}X$ complexes interfere with oligomerization of $MR^{wt}X$ complexes at DNA breaks, thus preventing efficient DNA repair *in vivo*.

(v) They used the structure of *Ct Rad50-DNA complex (5DAC)*. However, this is a coiled-coil truncated structure. Several studies proposed that the DNA interacts with coiled-coil region of *Rad50*. Xrs2 and *Mre11* also interacts with DNA, and thus the DNA-binding by the complex is not as simple as the authors have proposed.

Answer: We now mention in the revised manuscript that other MRX subunits/regions also mediate DNA binding, including the coiled-coils of *Rad50*.

4. *Oligomeric states of the MR complex are not well characterized. Authors claim that the wild-type MR dimers oligomerize into “non-aberrant aggregates”. If the MR dimers assemble into ordered forms of oligomers, what are their overall shape?*

Answer: It is important to note that the MR clusters formed in solution are not inactive, irreversible "aggregates". They can dissociate into MR dimers in the presence of DNA and subsequently form “pearls-on-a-string”-like structures (Fig. 1b and zoom-insets) and larger MR oligomers on DNA (Fig. 1d,ii) that are nucleolytically active (Fig. 1d,i). Yet, as discussed above, we note that these clusters are heterogeneous. We revised the text to clarify this point, avoiding the term "aggregate" (which would suggest inactive species) but also not describe the units as "ordered structures".

(i) Which forms of the oligomer enhance the endonuclease activity – as there are various oligomeric forms.

Answer: The MR proteins form a range of oligomer sizes, reaching an equilibrium of smaller and larger complexes depending on the conditions such as salt concentration, temperature and MR concentration (Fig. 1a and Supplementary Fig. 1b-d). As discussed above, we observed efficient DNA end resection under conditions when MR dimers bind DNA as “pearls-on-a-string” at lower MR:DNA ratios (Fig. 1b) or larger assemblies in which the individual MR dimers cannot be distinguished at higher MR:DNA ratios (Fig. 1d,ii). Especially the latter heterogeneous higher-order assemblies formed at higher MR:DNA ratios are highly nucleolytically active (Fig. 1d).

(ii) How various oligomers (including the worm-like lo MR oligomer) interact with pSae2?

Answer: We note that all tested MR variants (MR^{wt}, MR^{ho}, MR^{lo}) interact comparably with pSae2 in pulldown assays without DNA (Fig. 4i). We now included new data (Supplementary Fig. 4c), demonstrating that pSae2 indeed binds to MR^{wt} dimers assembled as “pearls-on-a-string” on DNA, resulting in larger assemblies in which the individual MR^{wt} dimers are obscured. However, where exactly pSae2 binds and how it activates MR remains to be further investigated.

REVIEWER COMMENTS

Reviewer #1 (Remarks to the Author):

The authors have strengthened the data and appropriately revised the manuscript. I can recommend publication and congratulate the authors for the nice work.

Reviewer #2 (Remarks to the Author):

This reviewer fully appreciates the authors' efforts to improve quality of the manuscript. Nevertheless, this reviewer has three major concerns that have not been clearly resolved in the revised manuscript.

First, I understand the resolution limits of negative staining and do not expect the quality of images at the level of single particle cryo-EM. Furthermore, this reviewer certainly does not think that the present study requires cryo EM analysis. However, I cannot agree on many interpretations in the present study. In line 120 and several other lines, authors have drawn a conclusion from several NS images in Fig 1 that the MR complexes laterally interact each other. Although authors have shown that MR dimers oligomerize through head to head interactions, I am not convinced on the authors' conclusion about the lateral interaction based on the images presented. This conclusion is also extended in Fig 2d. The 2 d i II (left NS image) showed three particles. Based on the image on top of the Fig 2d, scale bar and the last sentence in the section, I assume that each particle represents a cluster of a MR dimer. However, in the left of the image, authors draw a cartoon with three MR dimers laterally interact near the end. Does the image show three clusters of the MR dimers or three MR dimers? This cartoon (and others) could mislead readers by providing the impression that the MR dimer interacts laterally. Biochemical data support the role of β sheet in the dimer-dimer interface formation. However, the side of the β -sheet of a dimer could interact with any part of the Rad50 head. I also have difficulties in agreeing a pearls-on-a-string model on the images in Fig 1b ii and iii. For example, figure 1bii shows three clusters of MR complexes on top in addition to a zoomed cluster. Do authors claim all clusters show the pearls-on-a-string shape? I also examined carefully the attached figures in a response sheet, but not convinced. In fact, to this reviewer, Fig 1d ii showed an image close to pearls on a string, except each pearl contains a few MR dimers in this image.

Second, authors have justified that the ho mutant alleviates the electrostatic repulsive force (line 242 – 245 and 389-390) and Fig 3a. Authors showed that increased salt concentration disrupts the interactions between MR dimers, which suggests the electrostatic interaction are important forces for the dimer-dimer interface formation. However, I cannot agree on authors justification on the ho mutation. The only evidence authors claimed is that Glu125 is 4.8 Å apart from the another Glu125 of the crystallographic-symmetry related Rad50. Perhaps this led the authors idea on the lateral interaction. The two residues would be more apart if it is Asp in ScRad50. Yet, authors have proposed that the distance could be closer in solution. In the same logic, in solution water molecules may mediate the interactions. The crystal structure clearly reveals that Glu125 forms ion-pair with Arg126 of symmetry-related Rad50. Although authors briefly mentioned the presence of this ion-pair in the discussion, their overall rationale on the ho mutant is about the alleviation of the charge repulsion. Furthermore, they skipped the Glu125-Arg126 interaction in their figure in which most of other residues are fully drawn. This could totally mislead the readers.

Third, lines 327-329, 408-410. To justify that the lo quadruple mutation did not fully denature the intact structure, authors showed that overexpression of Rad50 lo on top of endogenous Rad50wt resulted in a pronounced growth defect and DNA damage sensitivity and claimed it as their strongest evidence. I do not think the mutation would completely destabilize the Rad50 structure to interfere complex formation. However, because at least one hydrophobic residue is fully buried and other two mutated residues are next to this residue, it is possible that simultaneous mutation of these residues disrupts the local structure around the two strands at the edge, which is distant from the Mre11

binding site and catalytic active site. Overexpressed lo mutant lacking endonuclease activity still can make a complex with endogenous Mre11 and the resulting complex impairs cell viability and response to DNA damage because of the defect of endonuclease activity rather than failed oligomerization.

Response to remaining reviewer comments (referee 2) for Kissling *et al.*

We would like to thank the reviewers for their interest and efforts. We are happy that both experts agree that the manuscript is of great importance and greatly improved during the revision process. However, while reviewer 1 has no further comments, reviewer 2 requested additional clarifications. A detailed point-by-point response how we addressed these points in the final figures and text follows below. The respective text changes are also highlighted in yellow in the final manuscript.

Reviewer #2 (Remarks to the Author):

This reviewer fully appreciates the authors' efforts to improve quality of the manuscript. Nevertheless, this reviewer has three major concerns that have not been clearly resolved in the revised manuscript.

First, I understand the resolution limits of negative staining and do not expect the quality of images at the level of single particle cryo-EM. Furthermore, this reviewer certainly does not think that the present study requires cryo EM analysis. However, I cannot agree on many interpretations in the present study. In line 120 and several other lines, authors have drawn a conclusion from several NS images in Fig 1 that the MR complexes laterally interact each other. Although authors have shown that MR dimers oligomerize through head to head interactions, I am not convinced on the authors' conclusion about the lateral interaction based on the images presented. This conclusion is also extended in Fig 2d. The 2 d i II (left NS image) showed three particles. Based on the image on top of the Fig 2d, scale bar and the last sentence in the section, I assume that each particle represents a cluster of a MR dimer. However, in the left of the image, authors draw a cartoon with three MR dimers laterally interact near the end. Does the image show three clusters of the MR dimers or three MR dimers? This cartoon (and others) could mislead readers by providing the impression that the MR dimer interacts laterally. Biochemical data support the role of b sheet in the dimer-dimer interface formation. However, the side of the b-sheet of a dimer could interact with any part of the Rad50 head. I also have difficulties in agreeing a pearls-on-a-string model on the images in Fig 1b ii and iii. For example, figure 1bii shows three clusters of MR complexes on top in addition to a zoomed cluster. Do authors claim all clusters show the pearls-on-a-string shape? I also examined carefully the attached figures in a response sheet, but not convinced. In fact, to this reviewer, Fig 1d ii showed an image close to pearls on a string, except each pearl contains a few MR dimers in this image.

We have revised the manuscript to indicate that (1) the head domain interactions may not necessarily be lateral, (2) that some, but not necessarily all MR molecules may form “pearls-on-a-string”-like structures, and (3) that the resolution of the EM images precludes determining the precise number of molecules in individual assemblies (e.g. in Fig. 2d). We would also like to point out that the addition of pSae2 to MR bound on DNA changes the morphology of the MR assemblies on DNA and the “pearls-on-a-string”-like arrangement of MR molecules is no longer visible due to the additional interactions (see Fig. 2d, Suppl. Fig. 4c).

Specifically, the changes include:

As suggested, we modified the cartoons in Fig. 2d and now show more than three MR dimers with pSae2 to avoid misleading the readers and to highlight that many MR-pSae2 molecules resect the 5' strand at the DNA end together to produce the observed single-stranded overhang. Moreover, we included the following sentence in the revised legend of Fig. 2d: "Due to the resolution of the negative staining images, the number of molecules or their orientation drawn in the cartoons may not exactly correspond to the assembly in the micrographs." A similar note was also included in the Supplementary file (Suppl. Fig. 4c). In addition, we added MR-pSae2 labels directly in the panels of Fig. 2d and Suppl. Fig. 4c for clarification.

Line 76: “We demonstrate that short-range DNA end resection is driven by MR oligomerization mediated by MR head domain interactions”. (i.e. we removed "lateral" from the statement).

Line 119: “The majority of MR molecules, however, were non-randomly distributed and oligomerized on DNA (Fig. 1b-ii, iii; red arrows). Some of these MR molecules appeared to form “pearls-on-a-string”-like assemblies on DNA, seemingly interacting with each other through the head domains, oftentimes with DNA loops in between (Fig. 1b-ii, iii zoom-insets and cartoons).” Line 141: “Interestingly, negative staining TEM analysis of the MR:DNA ratio with the highest resection activity in the nuclease assay (200 nM MR with 2.5 nM DNA, MR:DNA ratio 80:1; Fig. 1d-i) revealed that the MR dimers formed large oligomers on DNA, in which the individual MR dimers could no longer be easily distinguished (Fig. 1d-ii), in contrast to the assemblies with distinguishable MR dimers on DNA observed at lower MR:DNA ratios (Fig. 1b-ii, iii; Supplementary Fig. 1e).”

Accordingly, we have also adapted the legend of Fig. 1b-ii, iii and Fig. 1g, respectively: “MR dimers oligomerizing on DNA and some MR molecules forming “pearls-on-a-string”-like structures (zoom-insets of examples with cartoons), irrespective of DNA ends (Supplementary Fig. 1e,f).” “Model illustrating that in presence of DNA, MR oligomers dissociate to dimers which then oligomerize via head domain interactions at DNA ends or internally to “pearls-on-a-string”-like assemblies and larger clusters. Substrate compaction suggests DNA loop formation and coiled-coil tethering.”

Finally, the experiment in Fig. 2c shows that the observed resection at a DNA end occurs in a stepwise manner, compatible with the possible interpretation shown in the cartoons in Fig. 2d. We have revised the text (Line 187) to better highlight this finding: “Using shorter, 3'-labeled oligonucleotide-based DNA with one streptavidin-block at the 5'-terminated end revealed a gradual shortening of the 5'-strand by MR-pSae2 and the generation of DNA fragments of similar length (Fig. 2c). This indicates the sequential formation of several endonucleolytic incisions and the stepwise DNA end resection by many MR-pSae2 molecules that together produce the single-stranded overhang observed²¹. MR oligomerization may promote such joint DNA end resection²¹.”

Second, authors have justified that the ho mutant alleviates the electrostatic repulsive force (line 242 – 245 and 389-390) and Fig 3a. Authors showed that increased salt concentration disrupts the interactions between MR dimers, which suggests the electrostatic interaction are important forces for the dimer-dimer interface formation. However, I cannot agree on authors justification on the ho mutation. The only evidence authors claimed is that Glu125 is 4.8 Å apart from the another Glu125 of the crystallographic-symmetry related Rad50. Perhaps this led the authors idea on the lateral interaction. The two residues would be more apart if it is Asp in ScRad50. Yet, authors have proposed that the distance could be closer in solution. In the same logic, in solution water molecules may mediate the interactions. The crystal structure clearly reveals that Glu125 forms ion-pair with Arg126 of symmetry-related Rad50. Although authors briefly mentioned the presence of this ion-pair in the discussion, their overall rational on the lo mutant is about the alleviation of the charge repulsion. Furthermore, they skipped the Glu125-Arg126 interaction in their figure in which most of other residues are fully drawn. This could totally mislead the readers.

To avoid any misunderstanding, we revised the respective paragraphs and the legend of Fig. 3a. In the crystal structure, the Rad50 molecules are bound to separate DNA molecules, while on continuous DNA in solution the MR molecules could potentially sit closer together. To better describe this possibility, we changed the text as follows:

Line 222: “As a basis for mutational analysis, we employed a recent crystallographic structure showing a eukaryotic *C. thermophilum* (Ct) Rad50 dimer bound to a DNA molecule that forms a “quasi-continuous” structure with another Rad50 dimer-bound DNA molecule adjacent in the crystal, indicating a possible arrangement of two Rad50 dimers on DNA (Fig. 3a; RCSB Protein Data Bank PDB identifier: 5DAC⁴⁹). We hypothesized that an anti-parallel, two-stranded beta-sheet (black rectangle, β) and its connecting loop (black rectangle, L) protruding from the head domain of one Rad50 dimer in one asymmetric unit towards the identical structure of another Rad50 dimer in the adjacent

asymmetric unit of the crystal could represent a possible interaction site, especially on a continuous DNA molecule and in solution when MR molecules could come even closer on DNA than in the crystal structure shown.”

As requested, we adapted Fig. 3a to display CtArg126 and the interaction between CtGlu125 and CtArg126. Moreover, we revised the text and explicitly refer to the salt bridge between the CtGlu125-Arg126 residues (Lines 245/398 see below).

Accordingly, we have adjusted the legend of Fig. 3a as follows:

“Crystal structure of two *C. thermophilum* (Ct) Rad50 dimers bound to two separate oligonucleotide DNA molecules (PDB: 5DAC⁴⁹) used to generate a plausible interaction model as a guide for validation by mutagenesis. Possible interactions of adjacent Rad50 dimers on DNA *via* a small beta-sheet (β) and its connecting loop (L) protruding from the Rad50 head domain (rectangle), especially if the bound DNA was continuous. Zoom-inset shows red and blue residues mutated in the *S. cerevisiae* (Sc) Mre11-Rad50^{ho/lo} variants. The salt bridge between E125 and R126 residues of adjacent CtRad50 dimers is shown as grey dotted line (spacing 3.8 Å within the crystal). The repulsive E125 residues of adjacent CtRad50 dimers are spaced by 4.8 Å within the crystal (red dotted line) but could come closer on a continuous DNA molecule and in solution.”

To better address the points raised by reviewer 2, we have revised the text to further explain the rationale for the introduced mutations in the identified beta-sheet interaction:

Line 245: “Addressing the electrostatic component of MR oligomerization (Supplementary Fig. 1c), we substituted a polar residue in the potential interaction interface with an uncharged, non-polar alanine (Sc N121A, Ct N122) and removed a negative charge (Sc D124, Ct E125) to disrupt an attractive salt bridge between the D124 and R125 residues (Ct E125, R126) or to alleviate a potentially strong repulsive force between the aspartate residues D124 (Ct E125) of interacting ScRad50 molecules (red residues mutated; Fig. 3a zoom-inset and Fig. 3b; Rad50^{ho}).”

Our experimental analysis demonstrates that the Rad50^{ho} mutant exhibits beta-sheet dependent hyper-oligomerization. As in the absence of detailed structural data we can only speculate about the underlying molecular explanation, we have revised the text to better highlight the most likely explanation, while clearly mentioning the loss of the attractive salt bridge. The revised text now reads as follows:

Line 398: “As MR^{ho} displays enhanced oligomerization despite the loss of an attractive salt bridge between Sc D124 and R125 (Ct E125 and R126) due to the D124N mutation, we propose that the abolished charge repulsion of D124N residues between MR^{ho} molecules could drive this behavior. Although in the crystal structure model with two separate and only “quasi-continuous” DNA molecules (Fig. 3a) the Ct E125 (Sc D124) residues of adjacent Rad50 dimers are ~ 5 Å apart, they may be closer on a continuous DNA molecule in solution. Moreover, the hydrophobic residues in the beta-sheet could come closer when charge interactions are alleviated, which could explain the lower salt-sensitivity of MR^{ho} oligomers and the more pronounced beta-sheet signal in ThT staining. Yet, other head domain interactions are possible and high resolution structures of MR dimers adjacently bound on DNA will ultimately be needed to demonstrate the exact residues mediating beta-sheet-driven MR oligomerization.”

Third, lines 327-329, 408-410. To justify that the lo quadruple mutation did not fully denature the intact structure, authors showed that overexpression of Rad50 lo on top of endogenous Rad50wt resulted in a pronounced growth defect and DNA damage sensitivity and claimed it as their strongest evidence. I do not think the mutation would completely destabilize the Rad50 structure to interfere complex formation. However, because at least one hydrophobic residue is fully buried and other two mutated residues are next to this residue, it is possible that simultaneous mutation of these residues disrupts the local structure around the two strands at the edge, which is distant from the Mre11 binding site and catalytic active site. Overexpressed lo mutant lacking endonuclease activity still can make a complex

with endogenous Mre11 and the resulting complex impairs cell viability and response to DNA damage because of the defect of endonuclease activity rather than failed oligomerization.

The reviewer is correct that we cannot unambiguously exclude that the local structure is altered due to the introduced mutations and that this structural alteration contributes to the endonuclease activity defect of the MR^{lo} mutant apart from failed oligomerization. Accordingly, we revised the text and now explicitly mention this limitation:

Line 334: “We found that overexpression of Rad50^{lo} on top of endogenous Rad50^{wt} results in a pronounced growth defect and DNA damage sensitivity (Fig. 5d). Although we cannot exclude that the introduced mutations affect the endonuclease activity by altering the local structure, these results suggest that Rad50^{lo} retains the capacity to act in a dominant-negative manner, most likely by inhibiting oligomerization of the wild-type MRX pool.”

Line 411: “Moreover, Rad50^{lo} acts as a dominant-negative factor on endogenous Rad50^{wt} in overexpression experiments in yeast cells. Although we cannot exclude that the mutations in MR^{lo} negatively affect MR^{lo} activity beyond disrupting oligomerization, we note that oligomerization accompanies resection in MR^{wt}, and that the Rad50^{lo} mutant still binds pSae2, Mre11 and Xrs2 that mediates the observed nuclear localization in cells¹⁷. Moreover, MR^{lo} shows normal levels of exonuclease activity, which is inhibited by ATP γ S, as observed for wild-type MR²¹, indicating that the nuclease active site of MR^{lo} is sufficiently intact, given that the endo- and exonuclease activities of MR are mediated through the same active site⁵⁷.”

REVIEWERS' COMMENTS

Reviewer #2 (Remarks to the Author):

In the revised manuscript, authors clearly resolved the issues addressed by this reviewer and significantly improved the quality of a manuscript. This reviewer recommends the publication of the revised manuscript